# Optimizing Relevance Maps of Vision Transformers Improves Robustness

**Hila Chefer**    **Idan Schwartz**    **Lior Wolf**
School of Computer Science
Tel-Aviv University

## Abstract

It has been observed that visual classification models often rely mostly on spurious cues such as the image background, which hurts their robustness to distribution changes. To alleviate this shortcoming, we propose to monitor the model's relevancy signal and direct the model to base its prediction on the foreground object. This is done as a finetuning step, involving relatively few samples consisting of pairs of images and their associated foreground masks. Specifically, we encourage the model's relevancy map (i) to assign lower relevance to background regions, (ii) to consider as much information as possible from the foreground, and (iii) we encourage the decisions to have high confidence. When applied to Vision Transformer (ViT) models, a marked improvement in robustness to domain-shifts is observed. Moreover, the foreground masks can be obtained automatically, from a self-supervised variant of the ViT model itself; therefore no additional supervision is required. Our code is available at: `https://github.com/hila-chefer/RobustViT`.

## 1  Introduction

The reliance on simple image-level classification supervision, together with the sampling biases of object recognition datasets, leads to vision models that exhibit unintuitive behavior, as depicted in Fig 1. First, the models we tested (ViT [14], ViT AugReg [50], and DeiT [53]) tend to give disproportional high weight to the background of the image in the decision-making process. Second, the tested models occasionally regard a sparse subset of the pixels in the foreground object for the classification, disregarding much of the object's data. As argued by Geirhos et al. [19], and stated in [26] "image classification datasets contain 'spurious cues' or 'shortcuts' . For instance, cows tend to co-occur with green pastures, and even though the background is inessential to the identity of the object, models may predict 'cow', using primarily the green pasture background cue."

There is considerable evidence that context is a useful cue [31, 40]. However, many of the associated background elements and foreground shortcuts are only relevant to the specific data distribution, which leads to lack of robustness to distribution shifts [37, 43]. There are many methods for overcoming domain shifts, including domain adaptation techniques [16, 1] and methods that augment the training set or the training procedure [27, 38]. In this work, however, we opt for a direct approach, which monitors the relevancy score of the model for each image region, and manipulates the relevancy map to be focused on the regions within the foreground mask.

The method is based on a finetuning procedure, which is applied to a pretrained Vision Transformer (ViT) model. A relatively small set of samples, for which the foreground is given, is employed during this phase. In most of our experiments, we use three samples for half the classes, following work that examined the effect of transfer learning on half of the classes [60]. The ground-truth foreground mask is either human annotated [17], or estimated from a self-supervised ViT model [58].

36th Conference on Neural Information Processing Systems (NeurIPS 2022).

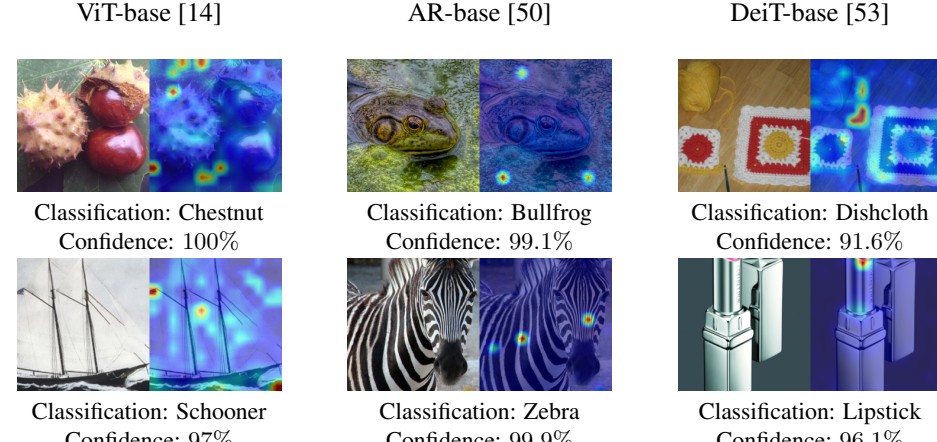

ViT-base [14]     AR-base [50]     DeiT-base [53]

Classification: Chestnut
Confidence: 100%

Classification: Bullfrog
Confidence: 99.1%

Classification: Dishcloth
Confidence: 91.6%

Classification: Schooner
Confidence: 97%

Classification: Zebra
Confidence: 99.9%

Classification: Lipstick
Confidence: 96.1%

Figure 1: Examples of the salient issues with ViTs. Each pair depicts an input image and its corresponding relevance map. The first row demonstrates examples of background-centered relevance, the second row shows examples of sparse foreground relevance. Both issues occur in all models-ViT [14], ViT AugReg [50] (AR), and DeiT [53], even if the confidence of the model is above 90%.

The finetuning procedure employs three loss terms. The first encourages the relevance map to assign lower values to the background of the image. The second term aims to remedy the sparse relevance issue by making larger parts of the foreground part of the relevancy map. The third term is a regularizer that ensures that the classification accuracy of the original model remains unimpaired.

Unsurprisingly, applying this method leads to a (modest) drop in accuracy on the original training dataset and on datasets with very similar distributions. In an extensive battery of experiments we show that (i) the classification accuracy on datasets from shifted domains increases considerably. This includes real-world unbiased and adversarial datasets, as well as synthetic ones that were created specifically to measure the robustness of the classification model, (ii) the resulting relevance maps demonstrate a significant improvement in focusing on the foreground of the image, i.e. the object, rather than on its background.

## 2 Related Work

Image classification datasets are becoming increasingly challenging, while at the same time models are growing more complex [12, 30, 51, 23]. With the rapid advancements in object recognition, the measuring stick is being narrowed down to a single number, accuracy [52]. By relying solely on accuracy, classifiers introduce biases, since they utilize shortcuts to select the right class [31, 19]. The shortcomings of models that rely on shortcuts have been demonstrated for domains other than vision, such as natural language processing [15] and multi-modal learning [18]. Therefore, it is becoming increasingly important to evaluate classifiers based on additional measures, such as robustness to distribution shifts, to ensure that the accuracy of the model does not conceal a problematic decision-making process.

**Detecting shortcuts** One way to assess the salient behavior of a model is to study Sufficient Input Subsets (SIS), i.e., the minimal number of pixels necessary for a confident prediction [7, 6]. Finding an SIS for a class can imply that the classifier has overinterpreted its input, since it can make a confident accurate decision using a small, sparse subset of pixels, which does not seem meaningful to humans. We study the SIS with gradients approach for ViT models, and find that it can be misleading. Specifically, SIS can be regarded as an adversarial method that can lead to high-confidence classification of *any label* from a sparse set of pixels, see Appendix A.

Explainability methods may be used to determine the reasons for the decisions made by classifiers. As an example, we may find that models tend to overlook objects with relevance maps (see Fig. 1). Gradients are a dominant and useful signal for model interpretation [45, 11, 34]. By adding input signals to the gradients, relevance maps were refined [21, 44, 48, 49]. Alternatives to gradient propagation for explanation include attribution propagation - a theory-driven method based on axioms, and permutation based on Shapley values [2, 35, 36, 44, 33]. For transformer architectures,

the combination of gradients and attention values has been shown to produce a viable interpretation of the model's prediction [8, 9].

**Reducing reliance on shortcuts**  Our method optimizes the relevancy maps of the model as a regularization term. Other works that investigated the use of relevancy to alleviate overfitting include Ross et al. [41], who introduced a regularization term for the input gradient, which reduces reliance on irrelevant cues, e.g., background pixels. Additional work in this vein has been conducted on medical data, studying how doctors classified a disease [46, 55]. Singh et al. [47] regularize feature representations of a category from its co-occurring context. Zhu et al. [61] enrich classifier representations by mimicking detection models. Importantly, unlike all the above methods, our method incorporates the foreground features and the classifier confidence, rather than considering only the background features. Furthermore, we apply our method not during training, but as a short finetuning process that is feasible for large models.

**Reducing bias**  A related yet slightly different task is that of debiasing models. One approach is training with adversarial strategies [28, 57]. Other approaches suggest generating counterfactual samples to mitigate reliance on biased features, e.g., masking gender features and preventing a captioning model from generating gender-related words [24, 62, 10]. Additionally, Kolesnikov et al. [29] examined background biases in object detectors.

## 3  Method

Our approach aims to direct vision models such that their decision will be based on the features of the object rather than on other supportive background features. To achieve this, we employ additional supervision to distinguish between the foreground and background features. The method finetunes the model in a way that encourages the class relevance map, obtained through a relevance computation method, to roughly resemble the segmentation map. This way, the decision-making process is focused on the foreground. The relevance map employed by our method is calculated using a recent advancement in explainability for transformer-based architectures [8]. A brief introduction of the explainability method used is provided in Appendix B.

The method employs a small set of labeled segmentation maps for distinguishing between the foreground and background of the input image. Our first loss term discourages the model from considering mostly the background:

$$\mathcal{L}_{\text{bg}} = \text{MSE}\left(\mathbf{R}(i) \odot \bar{\mathbf{S}}(i), 0\right), \tag{1}$$

where $i$ is the input image, $\mathbf{R}(i)$ is the relevance map produced for $i$, $\bar{\mathbf{S}}(i)$ is the inverse of the segmentation map for $i$, and $\odot$ is the Hadamard product. Put differently, $\mathcal{L}_{\text{bg}}$ extracts the relevance values assigned to the background using the provided segmentation, and encourages those values to be close to 0, which is the minimal possible relevance value.

Our second loss term encourages the model to consider as much information as possible from the foreground of the image:

$$\mathcal{L}_{\text{fg}} = \text{MSE}\left(\mathbf{R}(i) \odot \mathbf{S}(i), 1\right), \tag{2}$$

where $\mathbf{S}(i)$ is the foreground mask. This loss encourages the relevance of pixels inside the segmentation to be higher (1 is the maximal achievable relevance value). The overall explainability loss is constructed as follows:

$$\mathcal{L}_{\text{relevance}} = \lambda_{\text{bg}} \cdot \mathcal{L}_{\text{bg}} + \lambda_{\text{fg}} \cdot \mathcal{L}_{\text{fg}}, \tag{3}$$

where $\lambda_{\text{bg}}, \lambda_{\text{fg}}$ are hyperparameters. All our experiments apply the same choice of $\lambda_{\text{bg}} = 2, \lambda_{\text{fg}} = 0.3$. Note that the coefficient $\lambda_{\text{fg}}$ is much smaller than $\lambda_{\text{bg}}$. The reason is two-fold: (i) we find that the issue of overinterpreting the background is more common than the issue of partial relevance of foreground pixels, and (ii) $\mathcal{L}_{\text{fg}}$ implies a uniform relevance of 1 for all foreground pixels, which may be detrimental, as we wish to allow the model to be able to focus on specific features of the object.

Finally, in the absence of an additional regularization loss, the finetuning results in explanations that resemble the ground-truth segmentation, while the accuracy plummets due to the absence of encouragement to maintain high accuracy. Therefore, one must apply an additional loss term to ensure that the output distribution of the model remains similar to the original model. We opt to use a confidence-boosting loss for this purpose, which is constructed as follows:

$$\mathcal{L}_{\text{classification}} = \text{CE}\left(\mathcal{M}(i), \arg\max(\mathcal{M}(i))\right), \tag{4}$$

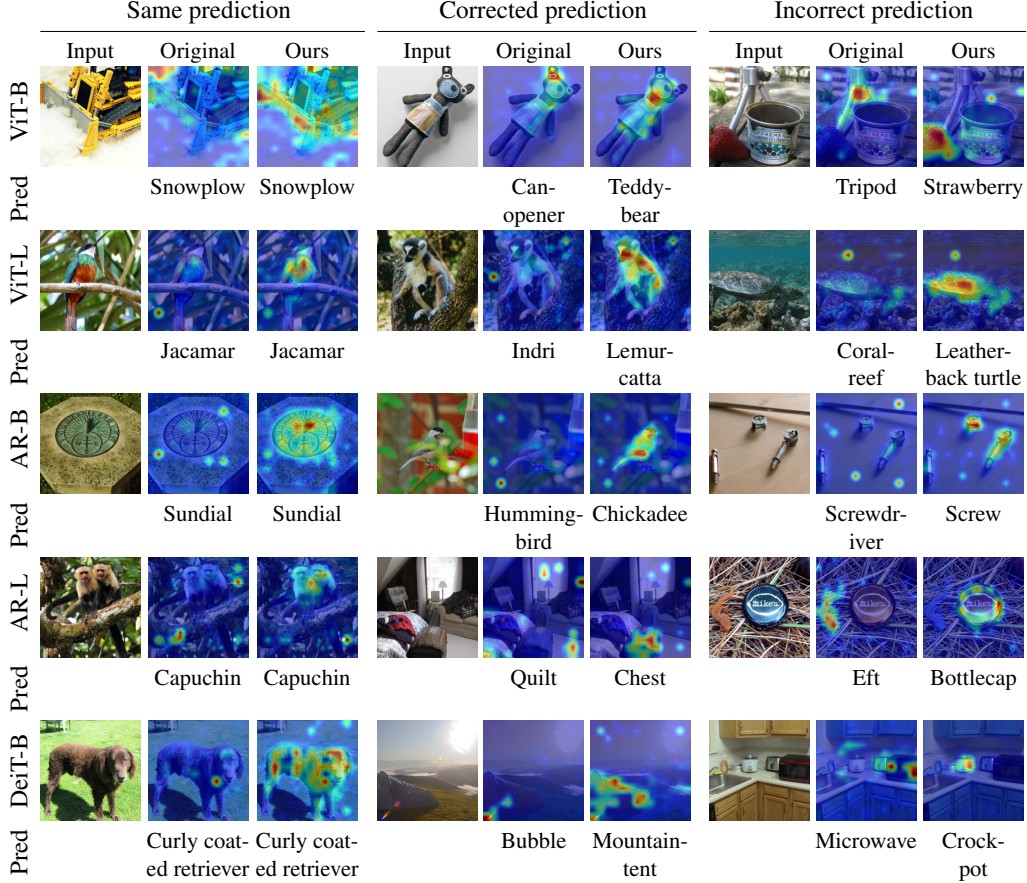

Figure 2: Examples from the ImageNet validation set of cases where our method does not change the prediction, corrects the prediction, and ruins the prediction. Even in cases where our method changes a correct prediction, there is often a rationale behind the modified prediction. The "Pred" row specifies the predictions before and after our finetuning. The examples are presented for the base, large models of ViT [14], ViT AugReg [50] (AR), and the base model of DeiT [53].

where $\mathcal{M}$ notates the vision model, and $\arg\max(\mathcal{M}(i))$ is the class predicted by $\mathcal{M}$ for the input image $i$. $\mathcal{L}_{\text{classification}}$ calculates the cross-entropy loss between the output distribution of $\mathcal{M}$ and the one-hot distribution where the predicted class is assigned a probability of $1$. In other words, this loss encourages the confidence of the predicted class to increase.

The overall loss for the finetuning process is, therefore:

$$\mathcal{L} = \lambda_{\text{relevance}} \cdot \mathcal{L}_{\text{relevance}} + \lambda_{\text{classification}} \cdot \mathcal{L}_{\text{classification}}, \tag{5}$$

where $\lambda_{\text{relevance}} = 0.8$, and $\lambda_{\text{classification}} = 0.2$ remain constant in all our experiments.

## 4   Experiments

The main hypothesis of this work is that improving the salient maps of ViTs trained on ImageNet will result in reduced overfitting, and better generalization to data from unseen distributions. We present a wide range of tests to confirm our hypothesis. Importantly, all the models we experiment on have been finetuned on ImageNet-1k, which is also the dataset we employ in our finetuning process.

First, we evaluate the improvement in robustness, i.e., the ability to maintain high accuracy under distribution shifts. The datasets with shifted distributions are only used for evaluation and contain both real-world datasets and synthetic ones. Second, we conduct segmentation tests following [9] to assess the effect of our method on the level of agreement between the relevancy maps and the foreground segmentation maps. Third, following [60], our method employs samples from a subset of

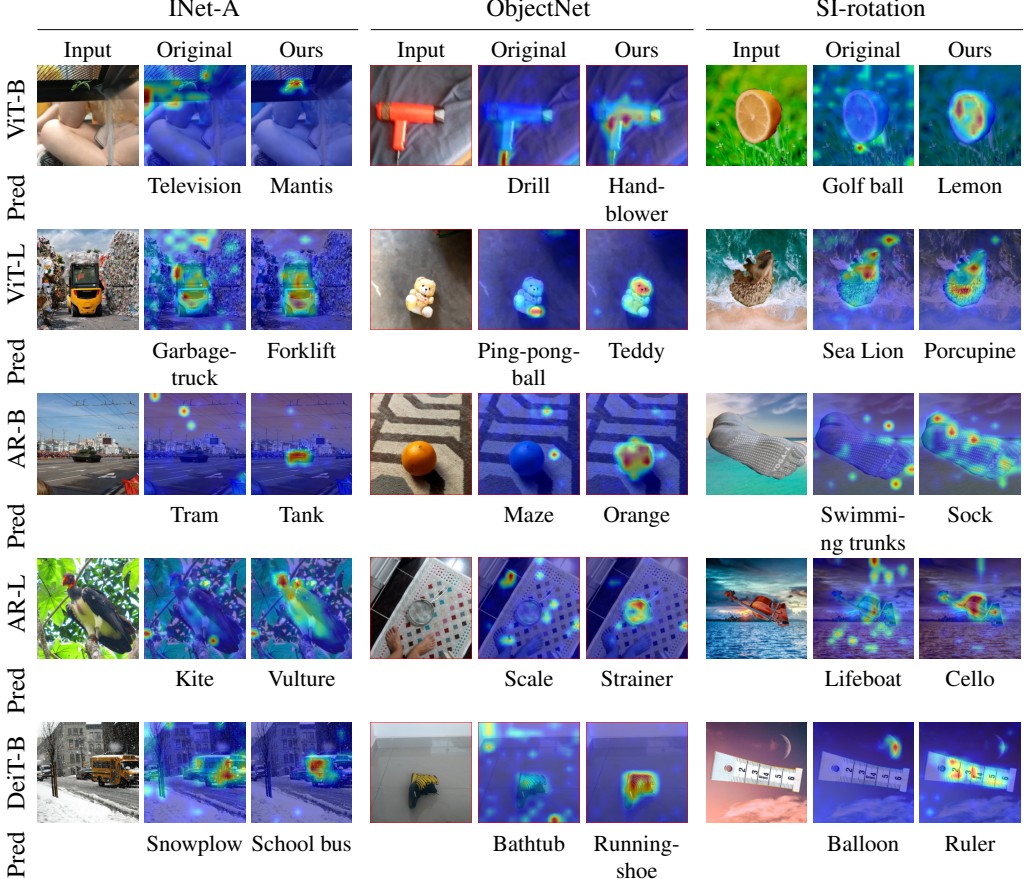

Figure 3: Examples of cases where our method corrects wrong predictions, alongside the original and modified (after finetuning) explainability maps. The "Pred" row specifies the predictions before and after our finetuning. The original classifiers focus on sparse or irrelevant data (e.g. the presence of snow leads DeiT-B to predict that a bus is a Snowplow, a porcupine is classified by ViT-L as a sea lion due to the presence of the ocean, a tank is classified by AR-B as a tram due to the presence of tram cables in the image, etc.). The examples are presented for the base, large models of ViT [14], ViT AugReg [50] (AR), and DeiT [53].

the classes during training, so that we can check whether the training classes (set A of the labels) differ from the set of classes not used during training (set B). Ideally, the method would have a positive effect on test samples from both sets A and B. Furthermore, in Appendix H we evaluate how sensitive the method is to the number of samples per class, and to the number of classes in set A.

**Additional experiments** Appendix L presents a comparison with self-supervised models and demonstrates that our method is beneficial even in a self-supervised setting. To validate the effectiveness of our method in improving the representations by ViTs, Appendix M contains $k-$NN experiments on datasets that contain non-ImageNet classes before and after applying our method.

**Robustness benchmarks** Several alternatives have been proposed to ImageNet [42] to examine robustness to distribution shifts. We conduct our experiments on ImageNet-v2 [39], ImageNet-A [26], ImageNet-R [25], ImageNet-Sketch [56], ObjectNet [4], and SI-Score [13]. A description of the datasets can be found in Appendix C.

**Baseline Methods** We focus on methods that resemble ours, i.e. methods that strive to correct overfitting by manipulating the saliency maps of the model. Our baselines include GradMask [46], and Right for the Right Reasons [41] (RRR). Both GradMask and RRR were originally applied during training; in this work, however, we focus on large vision models that require significant resources for training, which makes this approach computationally impossible for us. Therefore, we apply their accuracy and relevancy losses within a finetuning process, similar to ours, in order to test their success in a fair way, while adhering to computational limitations.

Both GradMask and RRR employ two loss functions. First, a classic cross-entropy loss with the ground-truth labels to ensure correct labeling, and second, a loss limiting the values of the gradients of irrelevant parts of the input. The latter resembles our background loss (Eq. 1), with gradients as the relevance map. We refer the reader to Appendix D for the full description of the applied losses.

We note that while using the gradient of the output w.r.t. the input is common practice for interpreting CNNs, these gradients are less stable for transformer-based models. For example, results presented in [32] demonstrate that for transformer-based models the classic Input×Gradient method violates faithfulness. We found it difficult to grid-search hyperparameters to fit both accuracy and relevancy losses simultaneously for the baselines. Furthermore, we had to tune the hyperparameters for each model separately to obtain an improved relevance loss. Our method, on the other hand, uses the same hyperparameter choice (see Sec. 3) for all models, which makes it far more stable to use, thus allowing us to run experiments on large models as well (i.e. ViT-L, ViT AugReg-L). We refer the reader to Appendix E for the full description of hyperparameters used in ours experiments.

**Models and Training**    To demonstrate the effectiveness of our method, we experiment on three types of ViT-based models: vanilla ViT [14], ViT AugReg (AR) [50], and DeiT [54] which presents techniques for efficiently training ViTs. For each model type, we experiment with different model sizes, in order to learn whether our method improves robustness across training techniques and model sizes. All models use $224$ resolution images with a patch size of $16 \times 16$. We use the implementation and pre-trained weights from [59]. The small, base models are finetuned on a single RTX 2080 Ti GPU, and the large models on a single Tesla V100 GPU. All models are finetuned as described in Sec. 3 for $50$ epochs, with a batch size of $8$. We use $3$ training images from $500$ ImageNet classes for our finetuning (overall- 1500 samples), and another $414$ images as a validation set. In most experiments, we simply select the first $500$ classes from ImageNet-S to be used for training (set A). For results with multiple random seeds and multiple choices of the $500$ classes to use for training, see Appendix F. The learning rate of each model is determined using a grid search between the values $5e - 7$ and $5e - 6$. As a rule of thumb, we select the highest learning rate for which the validation accuracy does not decrease by more than $2 - 3\%$ and $\mathcal{L}_{\text{classification}}$ does not increase. We find that our method is not sensitive to small shifts in the learning rate, thus this rule applies well to all models.

As mentioned in Sec. 3, we use segmentation maps to distinguish between the foreground and the background of an input image. Our experiments employ two options of obtaining segmentation maps for ImageNet training images. In the first option, we use human-annotated segmentation maps from the ImageNet-S dataset [17]. The ImageNet-S dataset contains $10$ training samples with their segmentation maps for $919$ of the ImageNet classes. We employ a considerably smaller subset, as detailed above. The second option does not employ extra supervision in the form of manually labeled images; instead, it employs the Tokencut object localization method presented in [58] to produce foreground segmentation maps.

**Results**    Tab. 1 presents the results of our method and the baseline methods applied to different ViT-based models. As can be seen, for real-world datasets, both adversarial datasets (INet-A) and datasets with random and controlled background, rotations, and view points (ObjectNet), our method significantly and consistently improves performance (average of $5.8\%, 5.0\%$ top-1 improvement on INet-A, ObjectNet, respectively). For datasets that contain art, sculptures, sketches etc. (INet-R, INet-Sketch) the increase in accuracy is less steep ($2.7\%, 0.9\%$ averaged top-1 improvement, respectively). This can be intuitively attributed to the fact that art and sketches often feature the object without a background, or with a uniform background. Additionally, as can be seen, although the baseline methods preserve accuracy on the datasets from the original ImageNet distribution (INet val set and INet-v2), they fall behind our method on real-world out-of-distribution datasets (INet-A and ObjectNet), indicating that the baselines are less successful in alleviating overfitting.

Following our method, there is a slight decrease in the performance of the models on data from the ImageNet distribution (INet val set and INet-v2), which can be attributed to the fact that we reduce some of the overfitting on the ImageNet data. Fig. 2 presents examples of all three possible prediction cases on the ImageNet validation set: cases where our method preserves the original classification, corrects the original prediction, and changes a correct prediction. As can be seen, while performance sometimes decreases with respect to the assigned label, in most cases the rationale of the class proposed by the modified model is clear, often indicating a more natural labeling option. In addition, in the cases where the prediction remains the same or is corrected, our method produces improved relevance maps.

Table 1: Robustness evaluation for ViT [14], ViT AugReg [50] (AR), and DeiT [54] with our method, and the baseline methods GradMask [46] and Right for the Right Reason (RRR) [41]. "Annotated segmentation" indicates whether we used annotated segmentation [17] or unsupervised localization [58]. "Original" stands for the model without finetuning. The bottom rows indicate the average change caused by our method across all architectures (on some models the baselines could not be run successfully; therefore, we do not compute their average change).

| Model | Method | Annotated segmentation | INet val R@1 | R@5 | INet-A R@1 | R@5 | INet-R R@1 | R@5 | Sketch R@1 | R@5 | INet-v2 R@1 | R@5 | ObjNet R@1 | R@5 |
|---|---|---|---|---|---|---|---|---|---|---|---|---|---|---|
| ViT-B | Original | ✗ | 81.5 | 96.0 | 16.0 | 37.0 | 33.8 | 48.5 | 35.4 | 57.4 | 71.1 | 89.9 | 35.1 | 56.4 |
| | GradMask | ✓ | 81.8 | 96.1 | 17.5 | 39.8 | 34.5 | 49.4 | 35.8 | 57.8 | **71.4** | **90.5** | 36.7 | 58.2 |
| | RRR | ✓ | **81.9** | **96.2** | 18.9 | 41.9 | 34.8 | 49.7 | 35.8 | 57.8 | **71.4** | **90.5** | 38.1 | 60.0 |
| | Ours | ✓ | 80.3 | 95.4 | **24.1** | **48.0** | **36.3** | **51.4** | 36.2 | **58.5** | 70.0 | 89.4 | **42.2** | **65.1** |
| | Ours | ✗ | 80.4 | 95.4 | 23.0 | 45.7 | 35.4 | 50.0 | 35.8 | 58.2 | 69.8 | 89.4 | 40.8 | 64.0 |
| ViT-L | Original | ✗ | **82.9** | **96.4** | 19.0 | 41.5 | 36.6 | 52.0 | 40.4 | 63.4 | 71.8 | 90.7 | 37.4 | 59.5 |
| | Ours | ✓ | 82.0 | 96.2 | **25.2** | 49.6 | 38.8 | 54.6 | 41.2 | 64.3 | 71.3 | 90.6 | 42.5 | 65.4 |
| | Ours | ✗ | 82.7 | **96.4** | **25.2** | **50.0** | **39.8** | **55.1** | **41.8** | **64.8** | **72.1** | **91.2** | **43.2** | **65.8** |
| AR-S | Original | ✗ | 81.4 | **96.1** | 13.0 | 33.9 | 31.2 | 47.1 | 32.8 | 54.2 | 69.9 | 90.1 | 34.3 | 55.8 |
| | GradMask | ✓ | 81.3 | **96.1** | 16.4 | 39.2 | 32.3 | 48.3 | 32.5 | 53.7 | 70.1 | **90.3** | 37.6 | 60.2 |
| | RRR | ✓ | **81.5** | **96.1** | 13.7 | 35.1 | 31.6 | 47.4 | 32.9 | 54.2 | **70.3** | 90.1 | 35.1 | 56.7 |
| | Ours | ✓ | 79.8 | 95.7 | 18.2 | 40.6 | **33.9** | **50.2** | 33.5 | 55.4 | 69.6 | 90.0 | 38.7 | 61.1 |
| | Ours | ✗ | 80.3 | 95.8 | **19.1** | **42.2** | 33.8 | 49.7 | **33.8** | **55.5** | 69.6 | 90.1 | **39.3** | **61.7** |
| AR-B | Original | ✗ | 84.4 | 97.2 | 23.9 | 49.2 | 41.0 | 57.8 | 43.1 | 65.7 | 73.8 | 92.3 | 41.4 | 63.7 |
| | GradMask | ✓ | 84.5 | **97.3** | 25.1 | 51.4 | 41.5 | 58.1 | 43.1 | 65.7 | 74.0 | **92.6** | 42.7 | 64.8 |
| | RRR | ✓ | **84.6** | **97.3** | 26.8 | 53.0 | 41.9 | 58.5 | 43.2 | 65.7 | **74.3** | **92.6** | 43.7 | 65.9 |
| | Ours | ✓ | 83.1 | 96.9 | **31.3** | 57.1 | **44.7** | **61.5** | 44.6 | **67.4** | 73.5 | 92.0 | **47.1** | **70.0** |
| | Ours | ✗ | 83.6 | 97.1 | 31.2 | **57.2** | 44.5 | 60.9 | **44.7** | **67.4** | 73.7 | 92.4 | 46.5 | 69.1 |
| AR-L | Original | ✗ | **85.6** | **97.8** | 34.7 | 61.0 | 48.8 | 64.9 | 51.8 | 73.6 | 75.8 | 93.4 | 46.5 | 68.3 |
| | Ours | ✓ | 85.1 | 97.5 | 42.1 | 67.5 | **54.0** | **69.1** | 54.2 | 75.8 | 75.8 | 93.4 | 51.6 | 73.2 |
| | Ours | ✗ | 85.4 | 97.6 | **42.4** | **68.0** | 53.8 | 69.0 | 54.1 | **75.8** | **76.1** | **93.6** | **52.0** | **73.5** |
| DeiT-S | Original | ✗ | 78.1 | 93.7 | 8.3 | 23.5 | 28.2 | 41.9 | 28.8 | 46.7 | 66.5 | 86.6 | 28.3 | 47.3 |
| | GradMask | ✓ | 77.0 | 93.6 | 7.9 | 24.7 | 26.6 | 40.5 | 26.0 | 43.5 | 64.5 | 85.6 | 28.2 | 48.6 |
| | RRR | ✓ | 78.1 | 94.1 | 9.0 | 26.9 | 26.9 | 40.6 | 26.9 | 44.4 | 66.0 | 86.7 | 29.3 | 49.9 |
| | Ours | ✓ | **78.6** | **94.5** | 10.1 | 29.0 | 29.3 | 43.6 | 29.1 | 47.8 | **67.3** | 87.3 | **31.6** | **53.0** |
| | Ours | ✗ | **78.6** | 94.4 | **11.0** | **30.3** | **29.9** | **44.4** | **29.4** | **48.0** | 67.1 | **87.4** | **31.6** | 52.9 |
| DeiT-B | Original | ✗ | 80.8 | 94.2 | 12.9 | 31.0 | 30.9 | 44.2 | 31.2 | 48.6 | **69.7** | 86.8 | 31.4 | 48.5 |
| | GradMask | ✓ | **81.1** | **95.3** | 15.1 | 36.9 | 31.0 | 45.5 | 31.2 | 49.1 | **69.7** | **88.7** | 33.5 | 53.1 |
| | RRR | ✓ | 81.0 | 95.2 | 14.8 | 37.0 | 30.7 | 45.1 | 30.9 | 48.8 | 69.5 | 88.6 | 33.6 | 53.3 |
| | Ours | ✓ | 80.5 | 94.9 | 17.2 | 40.0 | 32.4 | 47.0 | 30.9 | 49.2 | 69.1 | 88.3 | 35.9 | 56.2 |
| | Ours | ✗ | 80.5 | 95.0 | **18.3** | **40.9** | **32.8** | **47.5** | 31.5 | 49.9 | 69.3 | 88.5 | **36.3** | **56.6** |
| **Avg. change** | Ours | ✓ | -0.8 | 0.0 | +5.8 | +7.8 | +2.7 | +3.0 | +0.9 | +1.3 | -0.3 | +0.2 | +5.0 | +6.4 |
| | Ours | ✗ | -0.5 | 0.0 | +6.1 | +8.2 | +2.8 | +2.9 | +1.1 | +1.4 | -0.1 | +0.4 | +5.0 | +6.3 |

Additionally, as shown in Tab. 2 for the SI-Score dataset, which is a synthetic dataset designed for testing resilience for shifts in object locations, sizes, and rotations, there is a very steep improvement in performance across all models, while, once again, the baselines fall behind on all models and sizes.

Evidently, in both Tab. 1,2 our method works just as well and often better when using the unsupervised segmentation maps. This means that our method may be applied without requiring any manual supervision, except for the image label.

Fig. 3 presents example cases in which our method is able to correct the prediction of the original model on images from various robustness datasets. As can be seen, the original models tend to overinterpret the background, and therefore produce false classifications based on it. For example, a lemon is classified as a golf ball due to the grass in the background (third example in the first row), a tank is classified as a tram due to the tram cables at the top of the image (first example in the third row), and so on. Additional examples can be found in Appendix G.

Table 2: Robustness evaluation on the synthetic SI-Score dataset [13], which tests changes in object position, rotation, and size using our method and the baseline methods GradMask [46], Right for the Right Reasons (RRR) [41]. The models tested are ViT [14], ViT AugReg [50] (AR), and DeiT [54]. "Annotated segmentation" indicates if we used annotated segmentation [17] or unsupervised localization [58]. "Original" stands for the model without finetuning.

| Model | Method | Annotated segmentation | SI-location R@1 | SI-location R@5 | SI-rotation R@1 | SI-rotation R@5 | SI-size R@1 | SI-size R@5 |
|---|---|---|---|---|---|---|---|---|
| ViT-B | Original | ✗ | 33.3 | 52.2 | 39.1 | 58.3 | 55.6 | 76.2 |
| | GradMask | ✓ | 34.6 (+1.3) | 53.9 (+1.7) | 40.7 (+1.6) | 60.3 (+2.0) | 57.0 (+1.4) | 77.5 (+1.3) |
| | RRR | ✓ | 35.6 (+2.3) | 55.0 (+2.8) | 41.9 (+2.8) | 61.8 (+3.5) | 58.0 (+2.4) | 78.4 (+2.2) |
| | Ours | ✓ | **38.6** (+5.3) | **57.8** (+5.6) | **46.2** (+7.1) | **67.0** (+8.7) | **61.0** (+5.4) | **81.4** (+5.2) |
| | Ours | ✗ | 38.4 (+5.1) | 57.0 (+4.8) | 44.8 (+5.7) | 65.2 (+6.9) | 60.2 (+4.6) | 80.6 (+4.4) |
| ViT-L | Original | ✗ | 31.6 | 50.3 | 40.7 | 60.1 | 54.8 | 75.6 |
| | Ours | ✓ | 36.3 (+4.7) | 56.2 (+5.9) | **45.3** (+4.6) | 66.2 (+6.1) | 58.6 (+3.8) | 80.3 (+4.7) |
| | Ours | ✗ | **36.7** (+5.1) | **56.3** (+6.0) | **45.3** (+4.6) | **66.6** (+6.5) | **59.1** (+4.3) | **80.5** (+4.9) |
| AR-S | Original | ✗ | 32.4 | 51.7 | 40.6 | 59.6 | 55.4 | 75.7 |
| | GradMask | ✓ | 34.3 (+1.9) | 53.9 (+2.2) | 43.3 (+2.7) | 63.0 (+3.4) | 58.0 (+2.6) | 78.3 (+2.6) |
| | RRR | ✓ | 32.9 (+0.5) | 52.3 (+0.6) | 41.4 (+0.8) | 60.6 (+1.0) | 56.0 (+0.6) | 76.3 (+0.6) |
| | Ours | ✓ | **36.8** (+4.4) | **56.6** (+4.9) | **47.6** (+7.0) | **67.8** (+8.2) | **61.3** (+5.9) | **81.2** (+5.5) |
| | Ours | ✗ | 36.3 (+3.9) | 55.6 (+3.9) | 46.6 (+6.0) | 66.7 (+7.1) | 60.7 (+5.3) | 80.4 (+4.7) |
| AR-B | Original | ✗ | 40.5 | 60.8 | 48.1 | 68.3 | 60.6 | 80.4 |
| | GradMask | ✓ | 41.5 (+1.0) | 61.8 (+1.0) | 49.3 (+1.2) | 69.5 (+1.2) | 61.4 (+0.8) | 81.3 (+0.9) |
| | RRR | ✓ | 42.4 (+1.9) | 62.7 (+1.9) | 50.4 (+2.3) | 70.7 (+2.4) | 62.1 (+1.5) | 82.0 (+1.6) |
| | Ours | ✓ | 43.2 (+2.7) | 62.8 (+2.0) | 54.0 (+5.9) | 74.6 (+6.3) | 64.1 (+3.5) | 83.9 (+3.5) |
| | Ours | ✗ | **44.3** (+3.8) | **64.0** (+3.2) | **54.6** (+6.5) | **74.7** (+6.4) | **64.5** (+3.9) | **84.6** (+4.2) |
| AR-L | Original | ✗ | 43.8 | 64.2 | 52.4 | 72.5 | 62.3 | 82.2 |
| | Ours | ✓ | **48.3** (+4.5) | **68.5** (+4.3) | 57.0 (+4.6) | 77.2 (+4.7) | 66.4 (+4.1) | **86.0** (+3.8) |
| | Ours | ✗ | 47.4 (+3.6) | 67.4 (+3.2) | **58.0** (+5.6) | **78.1** (+5.6) | **66.5** (+4.2) | 85.6 (+3.4) |
| DeiT-S | Original | ✗ | 30.7 | 50.4 | 36.7 | 54.3 | 51.6 | 72.0 |
| | GradMask | ✓ | 32.0 (+1.3) | 50.7 (+0.3) | 38.9 (+2.2) | 56.7 (+2.4) | 54.1 (+2.5) | 74.0 (+2.0) |
| | RRR | ✓ | 32.0 (+1.3) | 51.0 (+0.6) | 38.5 (+1.8) | 56.3 (+2.0) | 53.9 (+2.3) | 73.8 (+1.8) |
| | Ours | ✓ | 32.3 (+1.6) | **51.5** (+1.1) | 40.6 (+3.9) | 59.4 (+5.1) | 55.8 (+4.2) | **76.3** (+4.3) |
| | Ours | ✗ | **32.5** (+1.8) | 51.4 (+1.0) | **41.0** (+4.3) | **59.6** (+5.3) | **56.0** (+4.4) | 76.1 (+4.1) |
| DeiT-B | Original | ✗ | 34.5 | 54.6 | 39.3 | 56.3 | 54.6 | 73.4 |
| | GradMask | ✓ | 34.1 (-0.4) | 54.9 (+0.3) | 39.1 (-0.2) | 58.3 (+2.0) | 55.2 (+0.6) | 75.8 (+2.4) |
| | RRR | ✓ | 34.4 (-0.1) | 55.2 (+0.6) | 40.4 (+1.1) | 58.5 (+2.2) | 55.3 (+0.7) | 75.8 (+2.4) |
| | Ours | ✓ | 36.6 (+2.1) | 57.0 (+2.4) | 42.9 (+3.6) | 61.5 (+5.2) | 58.0 (+3.4) | 78.2 (+4.8) |
| | Ours | ✗ | **37.8** (+3.3) | **58.1** (+3.5) | **44.2** (+4.9) | **62.7** (+6.4) | **59.3** (+4.7) | **79.0** (+5.6) |

Table 3: Evaluation of segmentation performance from relevance maps on the ImageNet-segmentation dataset [22] for ViT [14], ViT AugReg [50] (AR), and DeiT [54] before and after finetuning with our method. Metrics and dataset are taken from [9].

| Model | ViT-B Orig | ViT-B Ours | ViT-L Orig | ViT-L Ours | AR-S Orig | AR-S Ours | AR-B Orig | AR-B Ours | AR-L Orig | AR-L Ours | DeiT-S Orig | DeiT-S Ours | DeiT-B Orig | DeiT-B Ours |
|---|---|---|---|---|---|---|---|---|---|---|---|---|---|---|
| Pixel acc. | 76.3 | **82.1** | 73.4 | **82.5** | 76.7 | **83.3** | 76.6 | **81.2** | 65.2 | **78.9** | 78.7 | **80.8** | 79.0 | **81.3** |
| mIoU | 58.3 | **65.8** | 54.4 | **66.4** | 57.7 | **67.7** | 57.1 | **64.6** | 43.6 | **61.0** | 60.7 | **64.0** | 61.6 | **64.7** |
| mAP | 85.3 | **87.5** | 82.7 | **86.9** | 84.2 | **87.7** | 84.4 | **86.8** | 78.6 | **85.4** | 85.0 | **86.4** | 85.7 | **86.8** |

**Segmentation tests**   Since our motivation is to encourage the relevance to focus less on the background and more on as much of the foreground as possible, we test the resemblance of the resulting relevance maps to the segmentation maps following [9]. As can be seen in Tab. 3, our method significantly and consistently improves segmentation metrics on all models, indicating that our finetuning indeed achieves its goal. Appendix N contains perturbation tests that demonstrate that the improved relevancy maps after applying our method still faithfully reflect the model's reasoning.

**Comparing training classes to the other classes**   To ensure that the effects of our finetuning generalize to classes that were not included in the training set, we test the increase in robustness

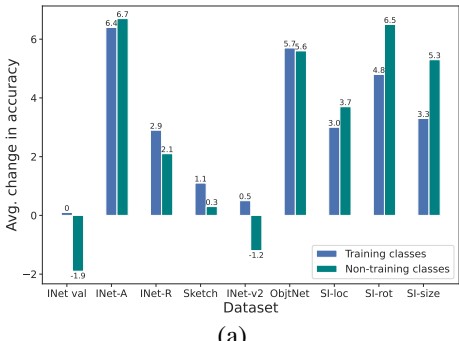
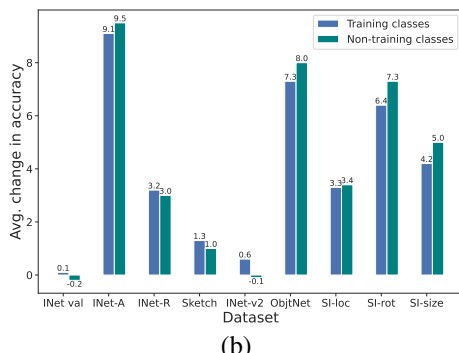

(a)                          (b)

Figure 4: Evaluation of the average change produced by our method on the training classes and on the classes that were not in the training set. Changes are averaged across the base models of ViT [14], ViT AugReg [50], and DeiT [53]. (a) top-1 average change, (b) top-5 average change.

Table 4: Ablation study for our method on ViT [14] and DeiT [53] base models. The table presents top-1 accuracy. We check the impact of each term of our loss on robustness, as well as the choice of confidence boosting versus cross-entropy with the ground-truth label (labeled as w/ ground-truth).

| Model | Method | INet val | INet-A | INet-R | Sketch | INet-v2 | ObjNet | SI-loc. | SI-rot. | SI-size |
|---|---|---|---|---|---|---|---|---|---|---|
| ViT-B | Original | 81.5 | 16.0 | 33.8 | 35.4 | 71.1 | 35.1 | 33.3 | 39.1 | 55.6 |
| | Ours | 80.3 | **24.1** | **36.3** | 36.2 | 70.0 | **42.2** | 38.6 | **46.2** | **61.0** |
| | w/o $\mathcal{L}_{\text{classification}}$ (Eq. 4) | 77.8 | 17.9 | 34.2 | 34.8 | 67.4 | 37.6 | 37.2 | 43.0 | 58.4 |
| | w/o $\mathcal{L}_{\text{bg}}$ (Eq. 1) | 81.5 | 19.1 | 35.9 | **36.4** | **71.4** | 39.7 | 34.9 | 42.3 | 58.4 |
| | w/o $\mathcal{L}_{\text{fg}}$ (Eq. 2) | 80.2 | **24.1** | 34.3 | 35.2 | 69.6 | 41.8 | **39.2** | 45.6 | 60.7 |
| | w/ ground-truth | **81.7** | 21.5 | 35.5 | 35.8 | 71.2 | 40.3 | 37.8 | 44.5 | 60.1 |
| | only $\mathcal{L}_{\text{bg}}$ | 66.0 | 18.0 | 23.2 | 25.8 | 55.1 | 32.1 | 33.3 | 35.0 | 50.5 |
| | only $\mathcal{L}_{\text{fg}}$ | 77.1 | 8.9 | 31.1 | 33.6 | 65.4 | 31.4 | 25.8 | 33.1 | 50.3 |
| DeiT-B | Original | 80.8 | 12.9 | 30.9 | 31.2 | 69.7 | 31.4 | 34.5 | 39.3 | 54.6 |
| | Ours | 80.5 | 17.2 | 32.4 | 30.9 | 69.1 | 35.9 | 36.6 | 42.9 | 58.0 |
| | w/o $\mathcal{L}_{\text{classification}}$ (Eq. 4) | **81.0** | 15.5 | **33.3** | **32.1** | **69.8** | **36.1** | **39.0** | **44.0** | **58.3** |
| | w/o $\mathcal{L}_{\text{bg}}$ (Eq. 1) | **81.0** | 13.2 | 30.5 | 30.2 | 69.6 | 33.0 | 32.7 | 39.7 | 54.5 |
| | w/o $\mathcal{L}_{\text{fg}}$ (Eq. 2) | 80.3 | **17.9** | 32.6 | 31.0 | 69.0 | 35.8 | 37.3 | 43.0 | 58.2 |
| | w/ ground-truth | 80.7 | 17.2 | 31.9 | 31.3 | 69.2 | 35.6 | 36.7 | 42.8 | 57.6 |
| | only $\mathcal{L}_{\text{bg}}$ | 71.7 | 13.1 | 25.3 | 24.7 | 59.7 | 29.2 | 38.4 | 37.4 | 53.4 |
| | only $\mathcal{L}_{\text{fg}}$ | 79.8 | 6.9 | 27.5 | 28.2 | 67.7 | 28.3 | 27.6 | 35.3 | 49.9 |

resulting from our method separately on the training classes and non-training classes. Fig. 4 presents the average improvement across the base models of ViT [14], ViT AugReg [50](AR), and DeiT [53]. As can be seen, both subsets of classes demonstrate a very similar increase in accuracy for the robustness datasets, with the classes that belong to the training set demonstrating better performance on the datasets from the original ImageNet distribution (INet val, INet-v2), as can be expected due to the fact that they were represented in the training set. The full results of this experiment are presented in Appendix I.

**Ablation Study** We conduct an ablation study to test the impact of each loss term on the result of the finetuning process. For brevity, Tab. 4 presents the top-1 accuracy results for each version of our method; the complementary table for top-5 accuracy can be found in Appendix J. Our ablation study is conducted on ViT-B and DeiT-B, as they demonstrate different advantages for each loss term.

First, we study the impact of removing each loss term (labeled "w/o $\mathcal{L}_{\text{classification}}$", "w/o $\mathcal{L}_{\text{bg}}$", "w/o $\mathcal{L}_{\text{fg}}$" in Tab. 4). As can be seen, validation accuracy is lost mostly due to our background loss (Eq. 1), as when we remove it, the accuracy remains intact, and yet, as we hypothesized, it is more effective than the foreground loss (Eq. 2) in increasing the robustness, since when removing it, the accuracy on the out-of-distribution datasets drops significantly. Additionally, the DeiT ablation demonstrates that in some cases, the classification loss does not contribute to the increase in robustness (other than for INet-A), nor is it necessary for preserving the original ImageNet validation accuracy. However, for ViT, the classification loss is crucial for avoiding a significant loss of accuracy. Next, we explore the variants where only one of the relevance loss terms $\mathcal{L}_{\text{bg}}, \mathcal{L}_{\text{fg}}$ is applied without the classification loss $\mathcal{L}_{\text{classification}}$ (labeled "only $\mathcal{L}_{\text{bg}}$", "only $\mathcal{L}_{\text{fg}}$" in Tab. 4). We note that by removing the classification

loss, the model is at risk of mode collapse since $\mathcal{L}_{bg}$ encourages the relevance on the background to be 0. The mode collapse, in this case, would be to zero-out the relevancy map for the entire image. In an analog manner, $\mathcal{L}_{fg}$ encourages a high relevance on the foreground and can cause a mode collapse where all the image receives uniform relevancy of 1. When applied together (the "w/o $\mathcal{L}_{classification}$ (Eq. 4)" ablation in Tab. 4) the two losses balance each other. However, when only employing one loss without the other and not adding a regularization term, the finetuning would encourage a mode collapse leading to lower accuracy.

Additionally, we perform an ablation to test the choice of confidence-boosting as the classification loss (Eq. 4), by replacing it with the classic cross-entropy loss with the ground-truth label. Our ablation study demonstrates the benefit of using confidence boosting over the ground-truth for the classification loss (labeled "w/ ground-truth" in Tab. 4). While the ground-truth variant preserves the original accuracy better (more so for ViT than for DeiT), using confidence boosting often improves robustness more significantly over using the ground-truth labels (see for example INet-A for ViT).

Finally, we refer the reader to Appendix J for additional experiments that explore different options to replace GAE [8] as the explainability algorithm to produce the relevance maps.

## 5 Discussion and limitations

A surge of works have explored the benefits of using large-scale datasets of unlabeled samples from the internet, and many techniques have been proposed to train vision models in an unsupervised or self-supervised manner [5, 20]. As a result, it is increasingly important to develop methods for boosting model robustness without requiring any labels. We note that our method is compatible with such frameworks. As can be seen from Tab. 1, 2, our method performs well, even when applied with foreground masks obtained in an unsupervised manner, using Tokencut. Additionally, all of our losses can be applied without knowledge of the ground-truth label, since our classification loss $\mathcal{L}_{classification}$ uses the predicted label, and the relevance maps could be propagated w.r.t. the predicted class.

Recently, it has been discovered that the effect of augmentation-based regularization is extremely uneven across classes [3]. While such a regularization improves performance on average, some classes benefit significantly from it, while other classes suffer from a large drop in accuracy. This variation is also apparent in the effect of our method. Above we have established that the method helps both classes which are included in the training set of the finetuning and classes which are not. However, in both sets there is considerable variability in the effect of individual classes. This makes sense, since some classes present more well-localized objects and some classes require more reliance on the context of the object. See Appendix K for these results.

## 6 Conclusions

Can segmentation information help image categorization? Intuitively, the answer has to be positive. However, despite some effort to manually segment images of classification datasets, the obtained improvement, if any, is soon overtaken by better image-level methods.

Here, we propose a generic way to improve classification accuracy that can be applied to virtually any image classifier. Applied to transformers, we present evidence to show that while the accuracy on the original dataset does not improve, there is an increase in accuracy for out-of-distribution test sets. The method optimizes the relevancy maps directly based on intuitive desiderata. It opens a new way for improving accuracy using explainability methods, which are currently seldom used for improving downstream tasks other than seeding weakly supervised segmentation methods.

## Acknowledgments

This project has received funding from the European Research Council (ERC) under the European Union's Horizon 2020 research and innovation programme (grant ERC CoG 725974). The first author is further supported by the Council for Higher Education in Israel. The contribution of the first author is part of a PhD thesis research conducted at Tel Aviv University.

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
