# Optimizing Relevance Maps of Vision Transformers Improves Robustness

**Hila Chefer**   **Idan Schwartz**   **Lior Wolf**
School of Computer Science
Tel-Aviv University

## A  Reinterpreting overinterpretation

One way to assess the salient behavior of a model is to study Sufficient Input Subsets (SIS), i.e., the minimal number of pixels necessary for a confident prediction [6]. A gradient signal can be utilized to find the sufficient pixels [5]. Finding an SIS for a class can imply that the classifier has overinterpreted its input since it can make a confident accurate decision using a small, sparse subset of pixels, which does not appear meaningful to humans.

We study the SIS with gradients approach for ViT models, and find that it can be misleading. Specifically, SIS can be regarded as an adversarial method that can lead to high-confidence classification of *any label* from a sparse set of pixels.

Fig. 1 demonstrates the resulting SIS pixels for $4$ randomly selected images from the ImageNet validation set, with $5$ randomly selected ImageNet labels- Gibbon, Black-widow, Common Iguana, Shovel, and Australian Terrier. As can be seen, we were able to find a subset of pixels for each random image and random label that made the classifier predict the random label with a confidence higher than $90\%$.

This can indicate that there is no guarantee that the classifier used the unnatural cues to decide for a given class based on the existence of a corresponding SIS, i.e. the subset of pixels considered an SIS is not necessarily indicative of the pixels used for the original decision, and resembles an adversarial attack in the sense that it can make the model predict unexpected outputs from a given input. Thus, we opt to use datasets designed specifically for testing robustness [11, 18, 13, 26, 3].

36th Conference on Neural Information Processing Systems (NeurIPS 2022).

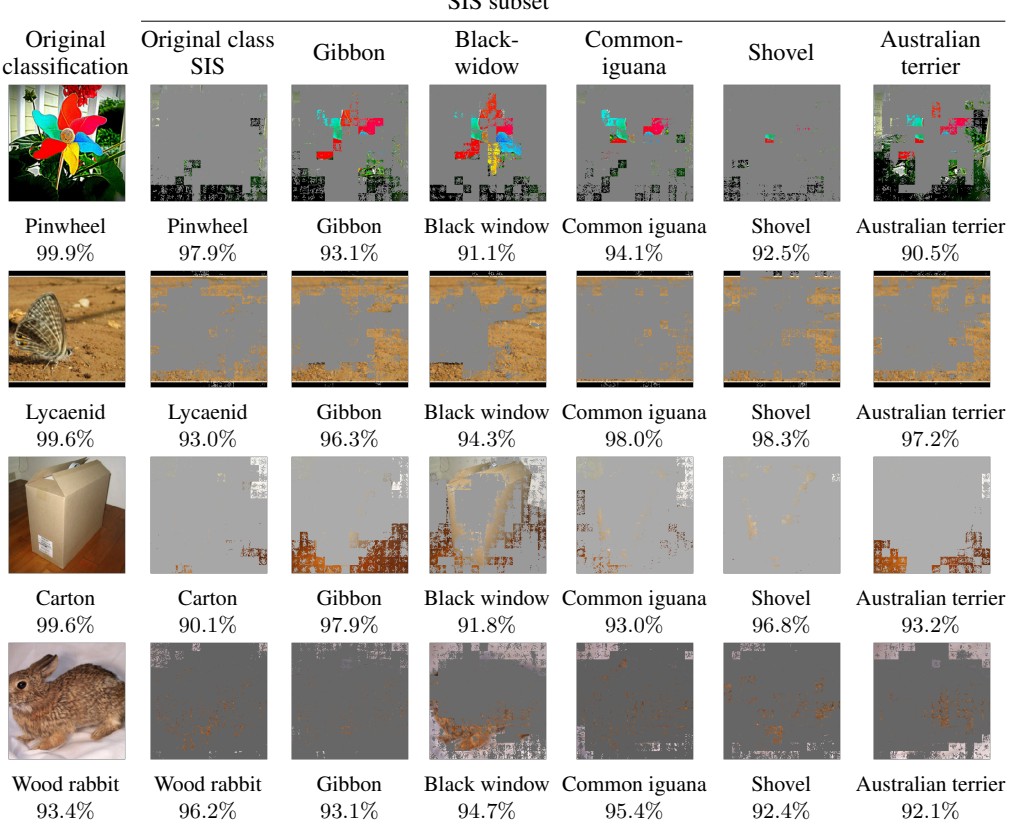

Figure 1: Examples of SIS subsets for 4 random images from the ImageNet validation set with 5 random ImageNet classes. As can be seen, an SIS subset can be found for all images with all classes with confidence higher than 90%, demonstrating that SIS subsets do not necessarily explain the prediction of the model.

# B   Background: explainability for Vision Transformers

Our method employs the Generic Attention Explainability (GAE) method [7] to create a relevance map for the input image tokens. In essence, GAE produces a relevance map $\bar{\mathbf{A}}$ for each self-attention layer. The final relevance map is the result of aggregating all the self-attention layers' relevance maps into a single map $\mathbf{R}(i)$ for the entire network in a forward pass using matrix multiplication.

We notate by $h$ the number of different attention heads, and by $s, d_h$ the number of tokens in the input sequence and the embedding dimension per head, respectively. Recall the self-attention mechanism employed by ViT:

$$\mathbf{A} = softmax \left( \frac{\mathbf{Q} \cdot \mathbf{K}^T}{\sqrt{d_h}} \right) \tag{1}$$

$$\mathbf{O} = \mathbf{A} \cdot \mathbf{V} \tag{2}$$

where $\mathbf{O} \in \mathbb{R}^{h \times s \times d_h}$ is the output of the attention module, $\mathbf{Q}, \mathbf{K}, \mathbf{V} \in \mathbb{R}^{h \times s \times d_h}$ are the query, key, and value for the attention mechanism, namely, different projections of the input to the self-attention module. $\mathbf{A} \in \mathbb{R}^{h \times s \times s}$ is the attention matrix, where row $i$ represents the attention coefficients of each token in the input with respect to the token $i$. The $softmax$ in Eq. 1 is applied, such that the sum of each row in each attention head of $\mathbf{A}$ is one.

Recall that after processing all the attention blocks, the ViT architecture regards only the classification token ([CLS] token) to produce the output distribution. Therefore, intuitively, the attention weights in the matrix $\mathbf{A}$ that correspond to the [CLS] token reflect the contribution of each token to the [CLS] token. Since the [CLS] token is the only one used for prediction, those contributions are often considered to be relevance values.

The authors in [7] follow this intuition to produce the relevance maps while handling two questions: (i) since, as demonstrated in [25], not all attention heads have an equal contribution to the prediction, how can the heads be averaged in a way that accounts for each head's relevance? (ii) given that a transformer includes multiple attention layers, how can the different attention layers be aggregated to a single attention matrix?

GAE tracks the updates by the attention of each layer and updates the relevance maps accordingly. In order to account for the varying roles of each head, gradients are used as weights for each attention head, similar to GradCAM [21] where gradients are used as weights for the features maps of the last CNN layer. To aggregate across attention layers, the method uses matrix multiplication and the identity matrix to account for residual connections as in attention Rollout [1].

Importantly, we chose to use GAE over pure-attention or variants of it such as attention Rollout since several papers on transformer explainability [7, 8, 17] have found pure attention weights to be unfaithful, i.e. pure attention does not loyally reflect the reasoning behind the decision-making process. The aforementioned works have also found GAE to be the most faithful method to interpret predictions by transformers. Additionally, due to the use of a combination of pure attention weights and attention gradients, GAE is easily derivable, therefore it is well-suited for our purposes of constructing a relevance-based loss.

The aggregated map is initialized using identity matrix: $\mathbf{R}(i) := I_{n_i}$. This is because before applying any attention, each token has no context, i.e., each token is self-contained, and has an influence of $1$ on itself. The relevance map of each self-attention layer of the Vision transformer is calculated as follows:

$$\nabla \mathbf{A} := \frac{\partial \mathcal{M}(i)}{\partial \mathbf{A}}; \qquad \bar{\mathbf{A}} = \mathbb{E}_h ((\nabla \mathbf{A} \odot \mathbf{A})^+), \tag{3}$$

where $\mathbf{A}$ is the attention map of the current layer, $\odot$ is the Hadamard product, $\mathcal{M}$ is the Vision transformer, $\mathcal{M}(i)$ is the logit that corresponds to the class we wish to visualize, and $\mathbb{E}_h$ is the mean across the heads dimension. Put differently, the method integrates the gradients of the desired output logit w.r.t. the attention map, in order to average across the attention heads and produce a single unified attention relevance map. The unified attention map $\bar{\mathbf{A}}$ is considered the relevance map of the attention layer.

Finally, to incorporate each layer's explainability map into the accumulated relevance maps the following propagation rule for self-attention layers is applied during a forward pass on the attention layers:

$$\mathbf{R}(i) \leftarrow \mathbf{R}(i) + \bar{\mathbf{A}} \cdot \mathbf{R}(i), \tag{4}$$

where $\bar{\mathbf{A}}$ is the attention relevance map for the self-attention layers, which is calculated using Eq. 3. To extract the relevance of each image token, the row of $\mathbf{R}(i)$ that corresponds to the $[\texttt{CLS}]$ token is used since the $[\texttt{CLS}]$ token alone determines the classification.

In this work, rather than using the relevance map as a form of explainability for a fixed model, we regularize the network to obtain the desired relevance map.

## C   Robustness benchmarks

Several alternatives have been proposed to ImageNet [20] to examine robustness to distribution shifts. We employ the following datasets in our experiments: (i) ImageNet-v2 [18], a new test set sampled from the same distribution, which reduces adaptive overfitting; (ii) ImageNet-A [13], a test set of natural adversarial samples; (iii) ImageNet-R [12] and ImageNet-Sketch [26], which contain renditions of objects (e.g., art, sculptures, sketches); (iv) ObjectNet [3], a real-world set with controls on object locations and view points, and (v) SI-Score [9], which is a synthetic dataset designed specifically for testing robustness to object location, rotation and size. We employ all the aforementioned datasets to evaluate accuracy when facing distribution shifts, and assess the model's resilience and ability to generalize.

## D   Baseline description

Both GradMask and RRR employ two loss functions. First, a classic cross-entropy loss with the ground-truth labels to ensure correct labeling, and second, gradients of the output of the model w.r.t. the input are used as a form of explanation, to limit the relevance scores of irrelevant parts of the input. This notion resembles our background loss (Eq. 1) with gradients as the relevance map.

In the following, we describe the loss terms applied in each method. For both methods, a standard cross-entropy loss with the ground truth class is applied to maintain accuracy, i.e.:

$$\mathcal{L}_{\text{classification}} = \text{CE}\left(\mathcal{M}(i), y_i\right), \tag{5}$$

where $i$ is the input image, and $y_i$ is a one-hot vector, where the ground truth classification of $i$ is assigned the value $1$.

GradMask applies a gradient-based loss to ensure that the gradients of the background are close to $0$:

$$\mathcal{L}_{\text{bg}} = \left\| \frac{\partial \hat{y}_i}{\partial i} \cdot \bar{\mathbf{S}}(i) \right\|_2, \tag{6}$$

where $\hat{y}_i$ is the predicted output for the ground-truth class, and $\bar{\mathbf{S}}(i)$ is, as before, the reversed segmentation map for the ground-truth class. Eq. 6 resembles our background loss (Eq. 1) with simple gradients w.r.t. the input image instead of the relevance map produced by GAE.

Similarly, Right for the Right Reasons (RRR) applies a loss to restrain the magnitude of explanations outside the relevant information. Their relevance loss is obtained as follows:

$$\mathcal{L}_{\text{bg}} = \left( \frac{\partial \sum_{k=1}^{K} \log(p_{i,k})}{\partial i} \cdot \bar{\mathbf{S}}(i) \right)^2, \tag{7}$$

where $k = 1, ..., K$ are the possible output classes, and $p_{i,k}$ is the probability assigned to the $k$-th class for image $i$ by the model.

For both GradMask and RRR, the following loss is used $\mathcal{L}_{\text{final}} = \lambda_{\text{bg}} \cdot \mathcal{L}_{\text{bg}} + \lambda_{\text{classification}} \cdot \mathcal{L}_{\text{classification}}$, where $\lambda_{\text{bg}}, \lambda_{\text{classification}}$ are hyperparameters. We note that while using the gradient of the output w.r.t. the input is common practice for interpreting CNNs, these gradients are less stable for transformer-based models. For example, results presented in [17] demonstrate that for transformer-based models the classic Input×Gradient method violates faithfulness.

In our experiments, we found it difficult to grid-search hyperparameters to fit $\mathcal{L}_{\text{bg}}$ and $\mathcal{L}_{\text{classification}}$. Furthermore, we had to tune $\lambda_{\text{bg}}, \lambda_{\text{classification}}$ for each model separately to obtain an improvement for $\mathcal{L}_{\text{bg}}$. Our method, on the other hand, uses the same hyperparameter choice (see Sec. 3), which makes it far more stable to use, thus allowing us to run experiments on large models as well. We refer the reader to Appendix E for the full description of hyperparameters used in our experiments.

# E Hyperparameters

In Tab. 1 we present the hyperparameter selection for all our experiments. Our method is stable and uses the same selection in all cases, other than the learning rate. The learning rates range from $6e - 7$ to $3e - 6$, allowing for a quick and easy grid search.

Tab. 2, 3 represent the hyperparameters of RRR, GradMask, respectively. For RRR, we had to tune the parameters per model, and the method is sensitive to the specific selection. For GradMask, we had to carefully tune the learning rate for each model. We found that the results of GradMask are sensitive to the specific learning rate selection, and to minor changes in the learning rate. We found it difficult to get the background loss to converge in both cases.

Table 1: Hyperparameter selection for our method for all models- ViT [10], ViT AugReg [23], and DeiT [24]. All hyperparameters are fixed except for the learning rate.

| Model | $\lambda_{\text{classification}}$ | $\lambda_{\text{relevance}}$ | $\lambda_{\text{bg}}$ | $\lambda_{\text{fg}}$ | Learning rate |
|---|---|---|---|---|---|
| ViT-B | 0.2 | 0.8 | 2 | 0.3 | $3e - 6$ |
| ViT-L | 0.2 | 0.8 | 2 | 0.3 | $9e - 7$ |
| AR-S | 0.2 | 0.8 | 2 | 0.3 | $2e - 6$ |
| AR-B | 0.2 | 0.8 | 2 | 0.3 | $6e - 7$ |
| AR-L | 0.2 | 0.8 | 2 | 0.3 | $9e - 7$ |
| DeiT-S | 0.2 | 0.8 | 2 | 0.3 | $1e - 6$ |
| DeiT-B | 0.2 | 0.8 | 2 | 0.3 | $8e - 7$ |

Table 2: Hyperparameter selection for the Right for the Right Reasons [19] method for all models- ViT [10], ViT AugReg [23], and DeiT [24]. Hyperapramters vary according to the model.

| Model | $\lambda_{\text{classification}}$ | $\lambda_{\text{bg}}$ | Learning rate |
|---|---|---|---|
| ViT-B | $2e - 6$ | $1e - 10$ | $2e - 6$ |
| AR-S | $2e - 8$ | $1e - 8$ | $1e - 5$ |
| AR-B | $2e - 7$ | $1e - 8$ | $5e - 7$ |
| DeiT-S | $2e - 6$ | $1e - 10$ | $1e - 5$ |
| DeiT-B | $2e - 6$ | $1e - 10$ | $5e - 6$ |

Table 3: Hyperparameter selection for the GradMask [22] method for all models- ViT [10], ViT AugReg [23], and DeiT [24].

| Model | $\lambda_{\text{classification}}$ | $\lambda_{\text{bg}}$ | Learning rate |
|-------|-----------------------------------|------------------------|---------------|
| ViT-B | $3e-9$ | 50 | 0.0002 |
| AR-S | $3e-9$ | 50 | 0.0005 |
| AR-B | $3e-9$ | 50 | $1e-5$ |
| DeiT-S | $3e-9$ | 50 | 0.005 |
| DeiT-B | $3e-9$ | 50 | 0.001 |

# F Random seed selection of the training classes

Tab. 4, and Tab. 5 present the results for the main experiment for multiple seeds, extending Tab. 1 and Tab. 2 of the main text. Each seed changes the 500 random classes used for finetuning with exactly the same hyperparameters.

As can be seen, the Standard Deviation is not large, especially in comparison to the performance gap. In all shifted-distribution datasets, the original result, before our intervention, is outside the standard error range for the multiple-seed experiments.

Table 4: Results using 3 random seeds. The table presents the average and standard deviation of the top-1 accuracy per dataset.

| Model | Method | INet val | INet-A | INet-R | Sketch | INet-v2 | ObjNet | SI-loc. | SI-rot. | SI-size |
|---|---|---|---|---|---|---|---|---|---|---|
| ViT-B | Original | **81.5** | 16.0 | 33.8 | 35.4 | **71.1** | 35.1 | 33.3 | 39.1 | 55.6 |
| | Ours | 80.3±0.1 | **23.6±0.7** | **36.1±0.3** | **36.3±0.1** | 70.1±0.3 | **41.7±0.6** | **38.6±0.2** | **46.1±0.4** | **60.9±0.4** |
| ViT-L | Original | **82.9** | 19.0 | 36.6 | 40.4 | **71.8** | 37.4 | 31.6 | 40.7 | 54.8 |
| | Ours | 82.0±0.1 | **25.0±0.3** | **38.6±0.4** | **41.1±0.1** | 71.2±0.2 | **42.6±0.2** | **36.6±0.4** | **45.3±0.4** | **58.6±0.5** |
| AR-S | Original | **81.4** | 13.0 | 31.2 | 32.8 | **69.9** | 34.3 | 32.4 | 40.6 | 55.4 |
| | Ours | 79.8±0.2 | **18.3±0.8** | **34.1±0.2** | **33.6±0.1** | 69.4±0.2 | **39.1±0.4** | **36.4±0.9** | **47.6±0.9** | **61.0±0.8** |
| AR-B | Original | **84.4** | 23.9 | 41.0 | 43.1 | **73.8** | 41.4 | 40.5 | 48.1 | 60.6 |
| | Ours | 83.0±0.1 | **30.9±0.3** | **45.3±0.7** | **44.9±0.4** | 73.3±0.2 | **47.0±0.6** | **43.8±0.7** | **54.7±0.6** | **64.6±0.5** |
| AR-L | Original | **85.6** | 34.7 | 48.8 | 51.8 | **75.8** | 46.5 | 43.8 | 52.4 | 62.3 |
| | Ours | 84.8±0.3 | **42.2±0.4** | **53.9±0.3** | **54.0±0.3** | 75.7±0.1 | **51.7±0.2** | **48.6±0.6** | **57.8±0.8** | **66.6±0.3** |
| DeiT-S | Original | 78.1 | 8.3 | 28.2 | 28.8 | 66.5 | 28.3 | 30.7 | 36.7 | 51.6 |
| | Ours | **78.7±0.1** | **10.4±0.3** | **29.3±0.1** | **29.0±0.2** | **67.3±0.3** | **31.5±0.1** | **32.2±0.2** | **40.6±0.4** | **55.6±0.4** |
| DeiT-B | Original | **80.8** | 12.9 | 30.9 | **31.2** | **69.7** | 31.4 | 34.5 | 39.3 | 54.6 |
| | Ours | 80.6±0.2 | **17.2±0.2** | **32.7±0.3** | 31.2±0.3 | 69.3±0.2 | **35.9±0.2** | **37.0±0.4** | **43.3±0.4** | **58.3±0.3** |

Table 5: Results using 3 random seeds. The table presents the average and standard deviation of the top-5 accuracy per dataset.

| Model | Method | INet val | INet-A | INet-R | Sketch | INet-v2 | ObjNet | SI-loc. | SI-rot. | SI-size |
|---|---|---|---|---|---|---|---|---|---|---|
| ViT-B | Original | **96.0** | 37.0 | 48.5 | 57.4 | **89.9** | 56.4 | 52.2 | 58.3 | 76.2 |
| | Ours | 95.4±0.0 | **47.1±0.8** | **51.3±0.2** | **58.7±0.3** | 89.4±0.1 | **64.7±0.5** | **57.5±0.3** | **66.6±0.5** | **81.2±0.3** |
| ViT-L | Original | **96.4** | 41.5 | 52.0 | 63.4 | **90.7** | 59.5 | 50.3 | 60.1 | 75.6 |
| | Ours | 96.2±0.0 | **49.2±0.6** | **54.6±0.3** | **64.3±0.1** | 90.6±0.3 | **65.5±0.2** | **56.4±0.3** | **66.3±0.5** | **80.3±0.3** |
| AR-S | Original | **96.1** | 33.9 | 47.1 | 54.2 | **90.1** | 55.8 | 51.7 | 59.6 | 75.7 |
| | Ours | 95.6±0.1 | **40.9±1.1** | **50.2±0.4** | **55.3±0.1** | 89.9±0.1 | **61.8±0.6** | **56.1±0.8** | **67.6±0.8** | **81.2±0.5** |
| AR-B | Original | **97.2** | 49.2 | 57.8 | 65.7 | **92.3** | 63.7 | 60.8 | 68.3 | 80.4 |
| | Ours | 96.8±0.1 | **56.8±0.3** | **62.0±0.7** | **67.9±0.6** | 91.9±0.1 | **69.9±0.5** | **63.5±0.8** | **75.1±0.6** | **84.4±0.5** |
| AR-L | Original | **97.8** | 61.0 | 64.9 | 73.6 | **93.4** | 68.3 | 64.2 | 72.5 | 82.2 |
| | Ours | 97.4±0.1 | **67.3±0.6** | **69.2±0.4** | **75.7±0.2** | 93.3±0.1 | **73.5±0.4** | **68.8±0.7** | **78.1±0.8** | **86.3±0.3** |
| DeiT-S | Original | 93.7 | 23.5 | 41.9 | 46.7 | 86.6 | 47.3 | 50.4 | 54.3 | 72.0 |
| | Ours | **94.5±0.1** | **29.0±0.5** | **43.6±0.0** | **47.6±0.4** | **87.4±0.1** | **53.0±0.1** | **51.3±0.2** | **59.3±0.5** | **76.1±0.3** |
| DeiT-B | Original | 94.2 | 31.0 | 44.2 | 48.6 | 86.8 | 48.5 | 54.6 | 56.3 | 73.4 |
| | Ours | **95.0±0.1** | **40.1±0.2** | **47.4±0.4** | **49.7±0.4** | **88.4±0.1** | **56.3±0.2** | **57.4±0.4** | **61.9±0.5** | **78.4±0.3** |

# G    Additional qualitative results

Fig. 2 provides more examples of cases where the salient behavior of the models prevents them from producing correct predictions. For example, an artichoke is classified as a green mamba due to partial consideration of the foreground (third example in the first row), a bagel is classified as a horse-chestnut due to the leaves in the background (third example in the third row), a racket is classified as a strainer due to the kitchen setting (second example in the second row), and a grasshopper is classified as a rock crab due to the rocks in the background (first example in the second row). By correcting the relevance maps, our method assists the models to achieve an accurate prediction.

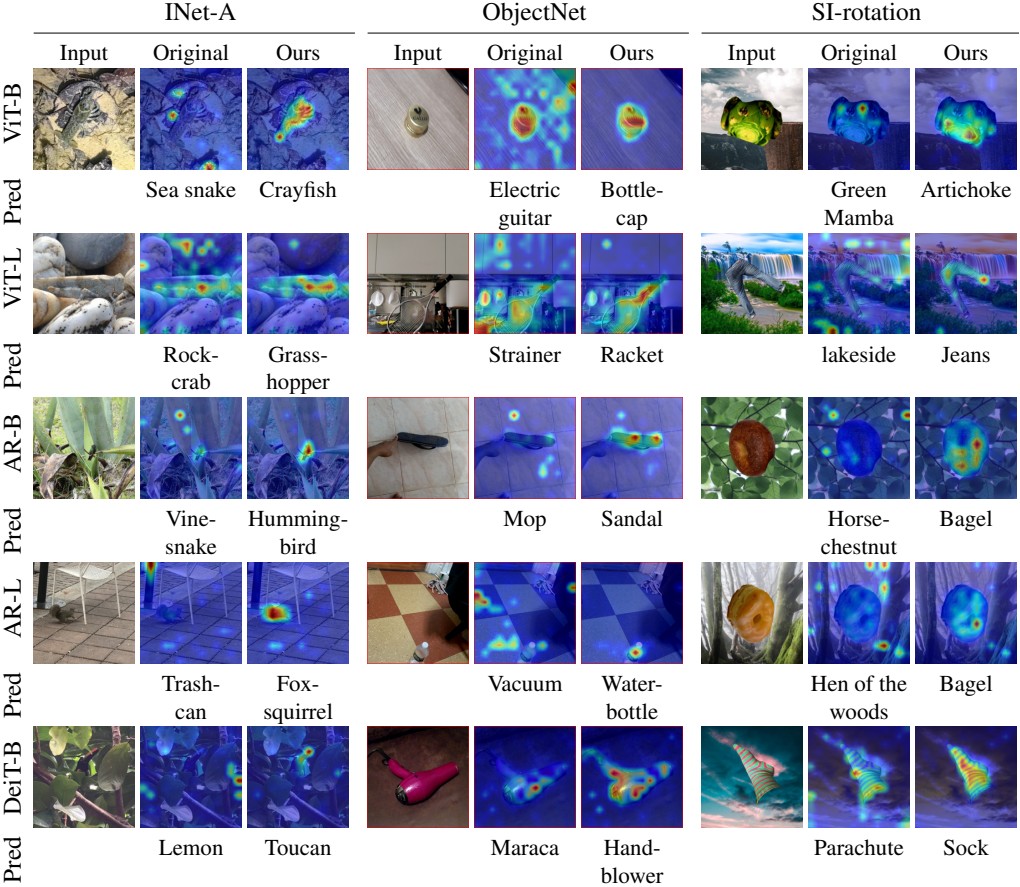

Figure 2: Examples of cases where our method corrects wrong predictions, alongside the original and modified (after finetuning) explainability maps (please zoom in for a better view). The "Pred" row specifies the predictions before and after our finetuning. The examples demonstrate cases where the original classifier relies on partial or irrelevant data, while our method rectifies the classification to be based on the object. The examples are presented for the large and base models of ViT [10], ViT AugReg [23] (AR), and the base model of DeiT [24].

# H Sensitivity tests

Fig. 3, presents sensitivity tests evaluating our robustness results on increasing number of samples per class (panel a), and increasing number of classes (panel b). Evidently, three samples for half the classes suffice to achieve the maximal improvement in robustness, while minimally harming the performance on ImageNet-based datasets (ImageNet val, ImagNet-v2). Using only two samples or considerably fewer classes does not harm performance much.

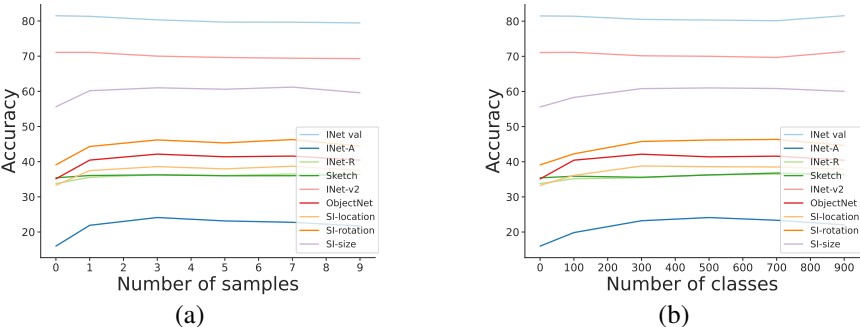

(a)                                                      (b)

Figure 3: Evaluation of our method's sensitivity to (a) the number of samples used per class in the training set (we use $n = 3$), and (b) number of training classes (we use $c = 500$), on ViT-B [10]. As can be seen, all the presented combinations of hyperparameters result in a significant increase in robustness, and a relatively modest decrease in INet val, INet-v2 accuracy.

# I Comparing training classes to the other classes

Tab. 6, 7 contain the full results of our experiment comparing the effect of our method on the classes in the training set and the classes outside of it. As can be seen, both the training and non-training classes benefit very similarly from applying our method when evaluated in terms of robustness on both real-world and synthetic datasets.

Table 6: Robustness evaluation on the classes that were included in the training set of our finetuning and classes that were not. The results are presented for the base models of ViT [10], ViT AugReg [23], and DeiT [24]. The last row indicates the average change for the training and non-training classes across the models for each dataset.

| Model | Train classes | | INet val R@1 R@5 | | INet-A R@1 R@5 | | INet-R R@1 R@5 | | Sketch R@1 R@5 | | INet-v2 R@1 R@5 | | ObjNet R@1 R@5 | |
|---|---|---|---|---|---|---|---|---|---|---|---|---|---|---|
| ViT-B | ✓ | Original | **82.0** | **96.0** | 14.8 | 35.2 | 32.9 | 47.4 | 36.8 | 58.4 | 71.3 | 89.8 | 36.0 | 55.8 |
| | | Ours | **82.0** | 95.7 | **22.9** | **46.1** | **35.5** | **50.5** | **38.1** | **59.7** | **71.4** | **89.9** | **43.5** | **64.6** |
| | ✗ | Original | **81.0** | **95.9** | 17.2 | 38.7 | 34.7 | 49.6 | 34.0 | 56.4 | **70.9** | **90.1** | 33.9 | 57.0 |
| | | Ours | 78.6 | 95.1 | **25.3** | **49.8** | **37.1** | **52.3** | **34.4** | **57.4** | 68.6 | 88.9 | **40.3** | **65.7** |
| AR-B | ✓ | Original | **85.0** | **97.3** | 22.6 | 46.7 | 42.3 | 58.7 | 41.5 | 64.9 | **73.3** | **92.4** | 42.6 | 62.7 |
| | | Ours | **85.0** | 97.2 | **30.3** | **54.9** | **44.8** | **62.2** | **42.0** | **66.1** | 72.1 | 91.9 | **48.4** | **69.2** |
| | ✗ | Original | **83.8** | **97.2** | 25.3 | 51.6 | 39.8 | 56.9 | 44.7 | 66.5 | 74.4 | **92.3** | 39.8 | 64.9 |
| | | Ours | 81.2 | 96.7 | **32.2** | **59.1** | **44.6** | **60.9** | **47.2** | **68.7** | **74.9** | 92.2 | **45.2** | **70.9** |
| DeiT-B | ✓ | Original | **81.7** | 94.4 | 11.7 | 29.4 | 29.6 | 42.9 | **32.0** | 49.3 | **70.3** | 86.5 | 32.7 | 48.8 |
| | | Ours | **81.7** | **95.2** | **15.2** | **37.5** | **30.9** | **45.5** | 31.5 | **49.6** | 69.4 | **88.4** | **36.6** | **55.3** |
| | ✗ | Original | **79.9** | 94.0 | 14.1 | 32.5 | 32.4 | 45.7 | **30.4** | 48.0 | **69.0** | 87.0 | 29.8 | 48.1 |
| | | Ours | 79.3 | **94.7** | **19.1** | **42.5** | **33.9** | **48.5** | 30.3 | **48.8** | 68.9 | **88.3** | **34.8** | **57.3** |
| **Avg. change** | ✓ | | 0 | +0.1 | +6.4 | +9.1 | +2.9 | +3.2 | +1.1 | +1.3 | +0.5 | +0.6 | +5.7 | +7.3 |
| | ✗ | | -1.9 | -0.2 | +6.7 | +9.5 | +2.1 | +3.0 | +0.3 | +1.0 | -1.2 | -0.1 | +5.6 | +8.0 |

Table 7: Robustness evaluation on the classes that were included in the training set of our finetuning and classes that were not for the synthetic datasets. The results are presented for the base models of ViT [10], ViT AugReg [23] (AR), and DeiT [24]. The last row indicates the average change for the training and non-training classes across the models for each dataset.

| Model | Train classes | | SI-loc. R@1 | R@5 | SI-rot. R@1 | R@5 | SI-size R@1 | R@5 |
|---|---|---|---|---|---|---|---|---|
| ViT-B | ✓ | Original | 34.2 | 52.5 | 40.5 | 60.3 | 56.5 | 76.1 |
|  |  | Ours | **38.6** | **58.0** | **46.3** | **68.1** | **60.4** | **80.4** |
|  | ✗ | Original | 32.0 | 51.8 | 37.1 | 55.6 | 54.4 | 76.3 |
|  |  | Ours | **38.6** | **57.5** | **46.0** | **65.6** | **61.9** | **82.9** |
| AR-B | ✓ | Original | 41.2 | 60.7 | 50.0 | 69.2 | 61.6 | 79.9 |
|  |  | Ours | **43.8** | **62.7** | **55.5** | **75.3** | **64.7** | **83.6** |
|  | ✗ | Original | 39.7 | 60.8 | 45.5 | 67.0 | 59.1 | 81.0 |
|  |  | Ours | **42.5** | **62.9** | **51.9** | **73.7** | **63.2** | **84.4** |
| DeiT-B | ✓ | Original | 35.2 | 54.2 | 40.3 | 56.3 | 56.1 | 72.3 |
|  |  | Ours | **37.1** | **56.6** | **43.5** | **61.6** | **59.0** | **77.0** |
|  | ✗ | Original | 33.6 | 55.1 | 38.0 | 56.1 | 52.5 | 74.8 |
|  |  | Ours | **35.8** | **57.5** | **42.1** | **61.4** | **56.7** | **79.7** |
| **Avg. change** | ✓ | | +3.0 | +3.3 | +4.8 | +6.4 | +3.3 | +4.2 |
|  | ✗ | | +3.7 | +3.4 | +6.5 | +7.3 | +5.3 | +5.0 |

## J  Ablation study

In Tab. 8 we conduct additional ablations to the experiments in Tab. 4 of the main text. We explore different options to replace GAE [7] as the explainability algorithm to produce the relevance maps. We consider two variants that employ pure attention weights. The first, labeled "Attention instead of [7]" in Tab. 4, utilizes the pure attention weights of the last attention layer as an explanation. The second, labeled "Rollout instead of [7]" in Tab. 4, employs attention Rollout [1] which combines the attention maps in all layers linearly using matrix multiplication. As can be seen, our method outperforms both these ablations, with the exception of ImageNet and ImageNet-v2. This can be explained by the experiments in [7, 8] which demonstrate that pure attention weights do not constitute a faithful explanation.

Tab. 9 presents the top-5 accuracy results of our ablation study to complement Tab. 4. As can be seen, the top-5 results are consistent with the top-1 results from Tab. 4.

Table 8: Additional ablation results for our method on ViT [10] and DeiT [24] base models. The table presents top-1 accuracy.

| Model | Method | INet val | INet-A | INet-R | Sketch | INet-v2 | ObjNet | SI-loc. | SI-rot. | SI-size |
|---|---|---|---|---|---|---|---|---|---|---|
| ViT-B | Original | 81.5 | 16.0 | 33.8 | 35.4 | 71.1 | 35.1 | 33.3 | 39.1 | 55.6 |
| | Ours | 80.3 | **24.1** | **36.3** | **36.2** | 70.0 | **42.2** | 38.6 | **46.2** | **61.0** |
| | Attention instead of [7] | 80.9 | 21.7 | 35.1 | 35.3 | 70.7 | 40.0 | 37.4 | 42.7 | 58.9 |
| | Rollout instead of [7] | 79.3 | 23.4 | 35.4 | 34.4 | 68.6 | 40.9 | 35.6 | 46.1 | 60.1 |
| DeiT-B | Original | 80.8 | 12.9 | 30.9 | 31.2 | 69.7 | 31.4 | 34.5 | 39.3 | 54.6 |
| | Ours | 80.5 | 17.2 | 32.4 | 30.9 | 69.1 | 35.9 | 36.6 | 42.9 | 58.0 |
| | Attention instead of [7] | **81.0** | 15.4 | 31.8 | 31.3 | **69.8** | 34.0 | 35.5 | 41.4 | 56.4 |
| | Rollout instead of [7] | 80.8 | 14.4 | 31.4 | 30.9 | 69.6 | 32.6 | 34.0 | 40.5 | 55.6 |

Table 9: Ablation study for our method on the ViT [10] and DeiT [24] base models. The table presents top-5 accuracy. We check the impact of each term of our loss on the robustness of the model, as well as the choice of confidence boosting versus cross-entropy with the ground-truth label (labeled as w/ ground-truth).

| Model | Method | INet val | INet-A | INet-R | Sketch | INet-v2 | ObjNet | SI-loc. | SI-rot. | SI-size |
|---|---|---|---|---|---|---|---|---|---|---|
| ViT-B | Original | 96.0 | 37.0 | 48.5 | 57.4 | 89.9 | 56.4 | 52.2 | 58.3 | 76.2 |
| | Ours | 95.4 | **48.0** | **51.4** | 58.5 | 89.4 | **65.1** | 57.8 | **67.0** | **81.4** |
| | w/o $\mathcal{L}_{classification}$ (Eq. 4) | 94.3 | 38.9 | 50.3 | 57.9 | 87.4 | 61.6 | 56.2 | 63.8 | 78.9 |
| | w/o $\mathcal{L}_{bg}$ (Eq. 1) | **96.1** | 41.8 | 50.9 | **58.5** | **90.5** | 62.1 | 54.1 | 62.7 | 78.9 |
| | w/o $\mathcal{L}_{fg}$ (Eq. 2) | 95.3 | 47.2 | 48.7 | 57.0 | 89.2 | 64.4 | **58.3** | 66.2 | 81.2 |
| | w/ ground-truth | **96.1** | 45.2 | 50.4 | 57.9 | 90.4 | 63.2 | 57.1 | 64.8 | 80.3 |
| | only $\mathcal{L}_{bg}$ | 85.1 | 35.9 | 34.9 | 45.2 | 75.9 | 53.2 | 49.4 | 52.6 | 70.9 |
| | only $\mathcal{L}_{fg}$ | 94.6 | 24.8 | 47.7 | 56.6 | 87.3 | 53.0 | 44.0 | 52.8 | 71.7 |
| | Attention instead of [7] | 95.7 | 44.8 | 50.0 | 57.3 | 89.9 | 62.5 | 56.1 | 62.5 | 79.0 |
| | Rollout instead of [7] | 94.9 | 47.1 | 50.0 | 55.8 | 88.7 | 63.8 | 53.9 | 66.6 | 80.6 |
| DeiT-B | Original | 94.2 | 31.0 | 44.2 | 48.6 | 86.8 | 48.5 | 54.6 | 56.3 | 73.4 |
| | Ours | 94.9 | 40.0 | 47.0 | 49.2 | 88.3 | 56.2 | 57.0 | 61.5 | 78.2 |
| | w/o $\mathcal{L}_{classification}$ (Eq. 4) | **95.2** | 37.4 | **48.4** | **51.4** | **88.9** | 56.8 | **60.1** | **63.1** | **79.5** |
| | w/o $\mathcal{L}_{bg}$ (Eq. 1) | **95.2** | 34.4 | 44.7 | 47.9 | 88.3 | 52.4 | 53.2 | 57.5 | 74.7 |
| | w/o $\mathcal{L}_{fg}$ (Eq. 2) | 94.9 | **40.1** | 47.2 | 49.3 | 88.1 | 56.4 | 57.6 | 61.4 | 78.5 |
| | w/ ground-truth | 95.0 | 39.6 | 46.5 | 49.6 | 88.2 | 55.9 | 57.3 | 61.2 | 78.0 |
| | only $\mathcal{L}_{bg}$ | 85.9 | 26.3 | 35.4 | 39.7 | 75.6 | 44.1 | 55.0 | 51.2 | 70.2 |
| | only $\mathcal{L}_{fg}$ | 94.8 | 22.7 | 42.2 | 46.2 | 87.4 | 47.0 | 46.9 | 53.2 | 71.1 |
| | Attention instead of [7] | **95.2** | 37.5 | 46.2 | 49.4 | 88.7 | 53.5 | 56.5 | 59.7 | 76.7 |
| | Rollout instead of [7] | **95.2** | 35.7 | 45.7 | 48.6 | 88.4 | 52.5 | 54.7 | 58.4 | 76.0 |

## K    Per class robustness analysis

There is considerable variability in the effect of our method between the different classes. This is similar to the varying effect of regularization reported by Balestriero et al. [2] and it is reasonable to assume that it is a common phenomenon among regularization techniques.

Fig. 4-10 depict the classes that benefited the most by our finetune process on ViT-B and those that were harmed the most. For datasets with $1,000$ classes, we present the $50$ classes with the most beneficial or harmful change for readability. Other datasets are presented with all their classes.

Inspecting the classes with the largest amount of change, we can observe a few trends. In some cases, classes with relatively small objects considerably benefit, and classes with objects that either reside in the background or benefit from the context are harmed, as can be expected. In some cases, classes with a small number of test samples, regardless of the type of object, lead to a higher absolute change, due to statistical reasons.

In order to study the limitations of our method, we inspect the classes where it fails the most for INet-A and INet-V2. These two test sets represent different amounts of domain shift from the original ImageNet (INet-V2 is very close to ImageNet, while INet-A is out-of-distribution). As shown in the paper, our method increases robustness to domain shifts.

Fig. 11 depicts the two classes, out of each of the two test sets (INet-A and INet-V2) that were harmed the most: pool table and breastplate for INet-A and breastplate and miniature poodle for INet-v2. From each class, we show the effect of our method on the first three samples in which a correct classification before finetuning turned into a wrong classification (so we avoid cherry-picking).

The pool table samples tend to show other objects in the foreground. In the top-left sample, the object is identified as a bucket (it is a hat, which is the 2nd top prediction). In the other two samples for pool tables and INet-A, the foreground object is correctly identified. In all three cases, the heatmap explains the result. In the breastplate case, there are only 11 test samples, two of which were wrongly classified by our method as other, reasonable classes.

For INet-v2, following our method, some breastplates are identified as cuirass (oxford dictionary: a piece of armor consisting of breastplate and backplate fastened together), which is a similar class. In the 2nd most affected class, miniature poodles are identified as toy poodles, or as a Lhasa Apso, which are similar dog breeds.

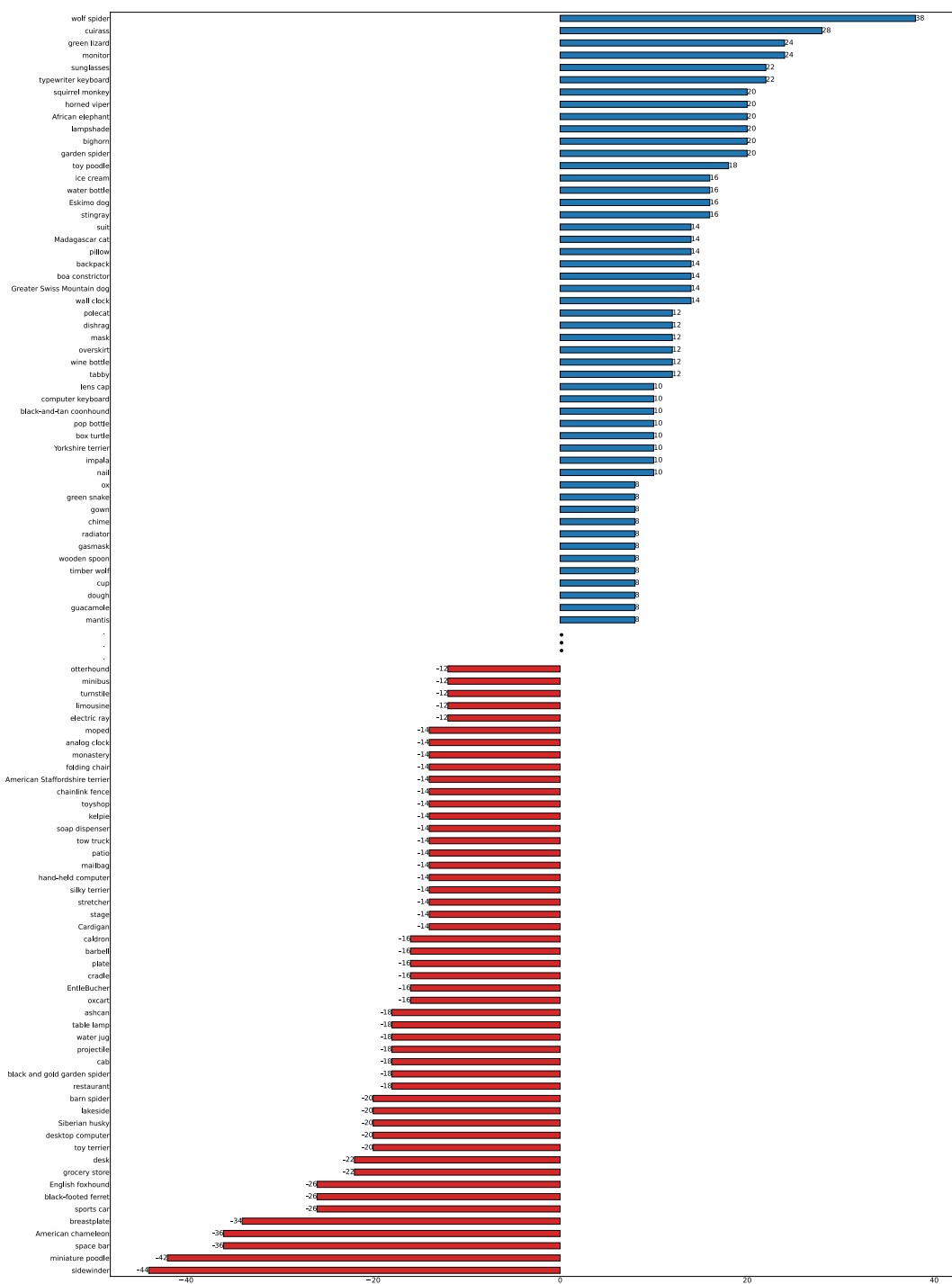

Figure 4: The 50 classes which most benefited and the 50 classes that were most harmed by the finetune procedure for ImageNet.

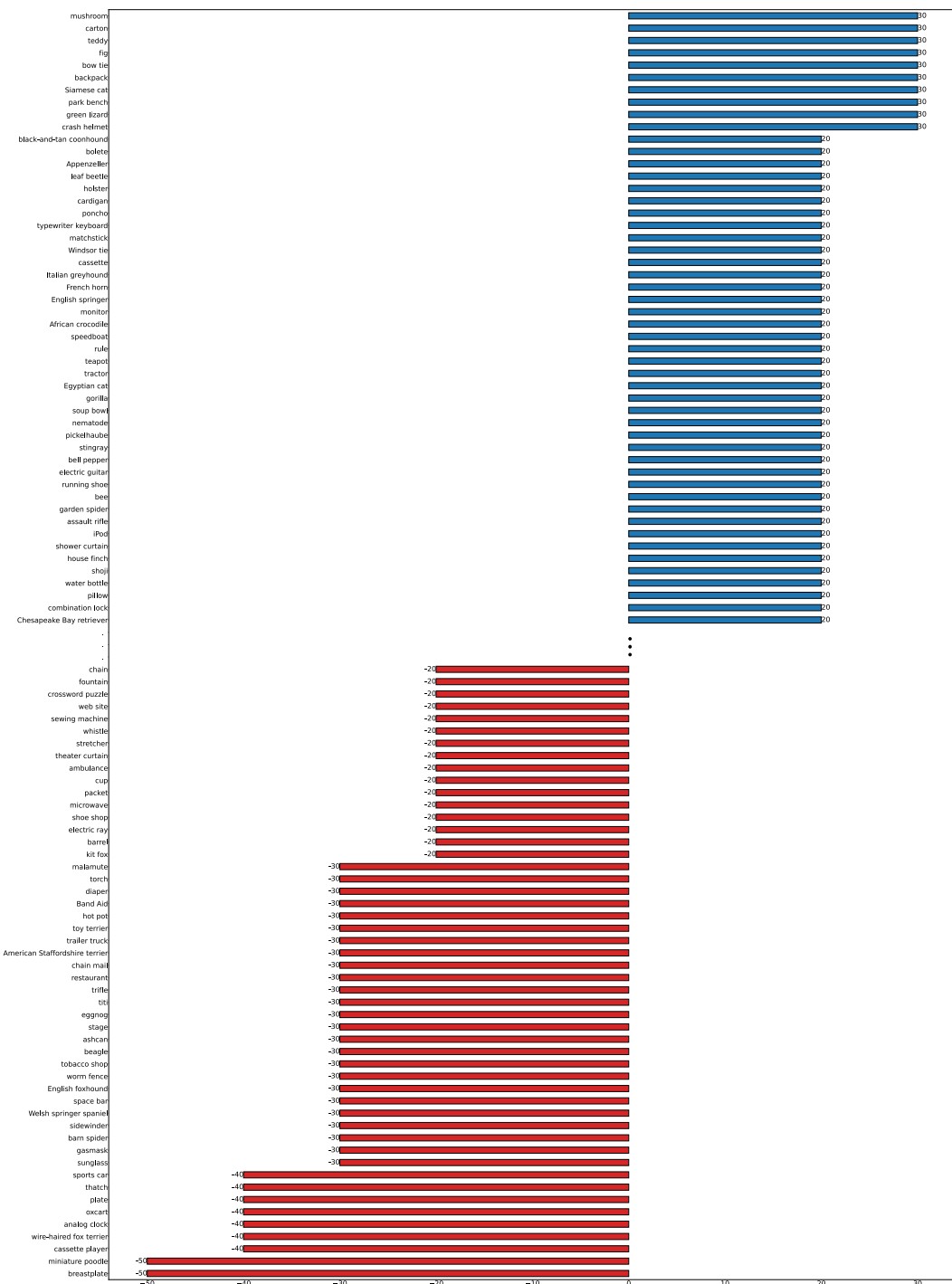

Figure 5: The 50 classes which most benefited and the 50 classes that were most harmed by the finetune procedure for ImageNet-v2.

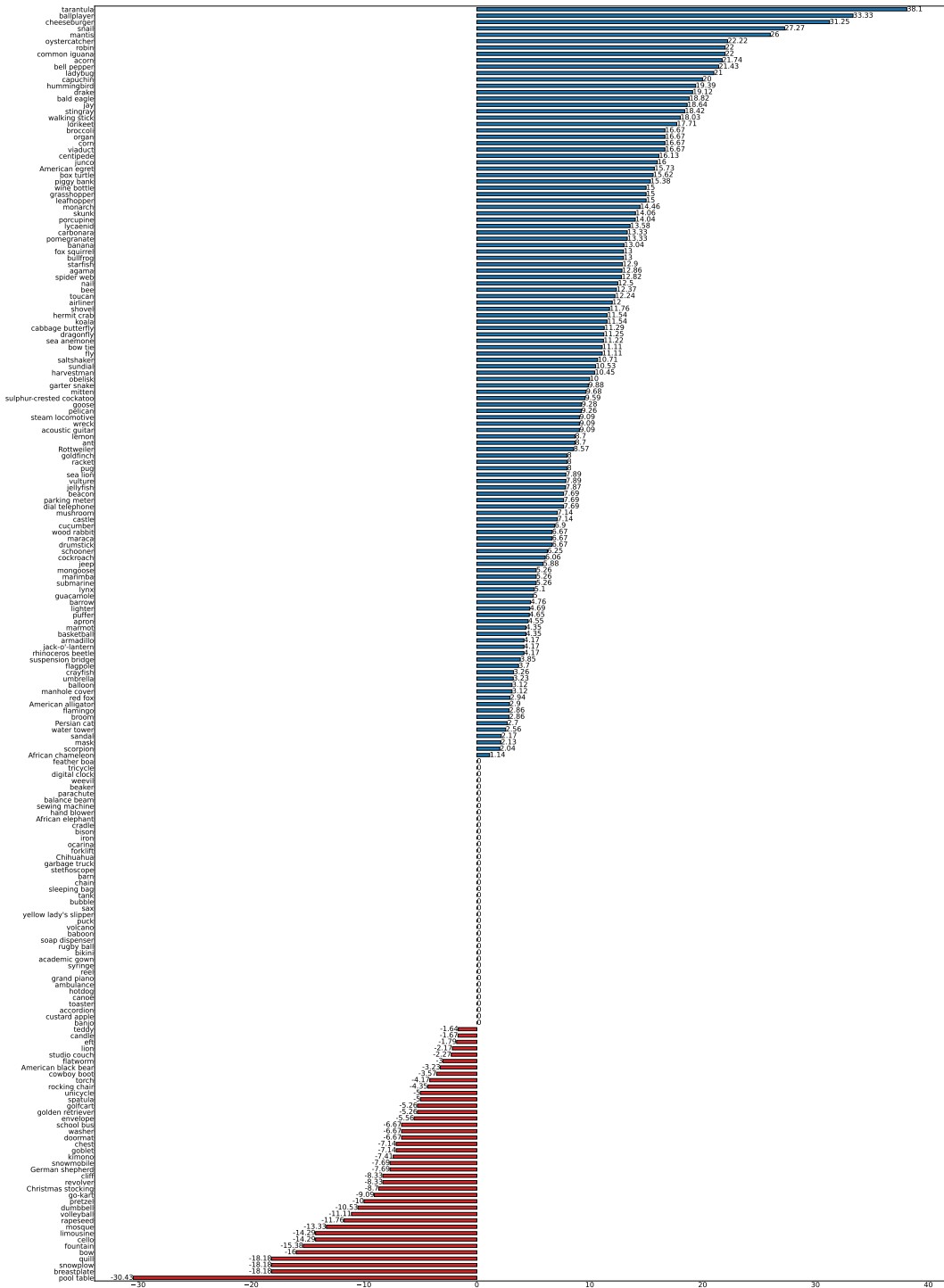

Figure 6: The effect of the finetune procedure on each class for ImageNet-A (there are some classes with zero effect).

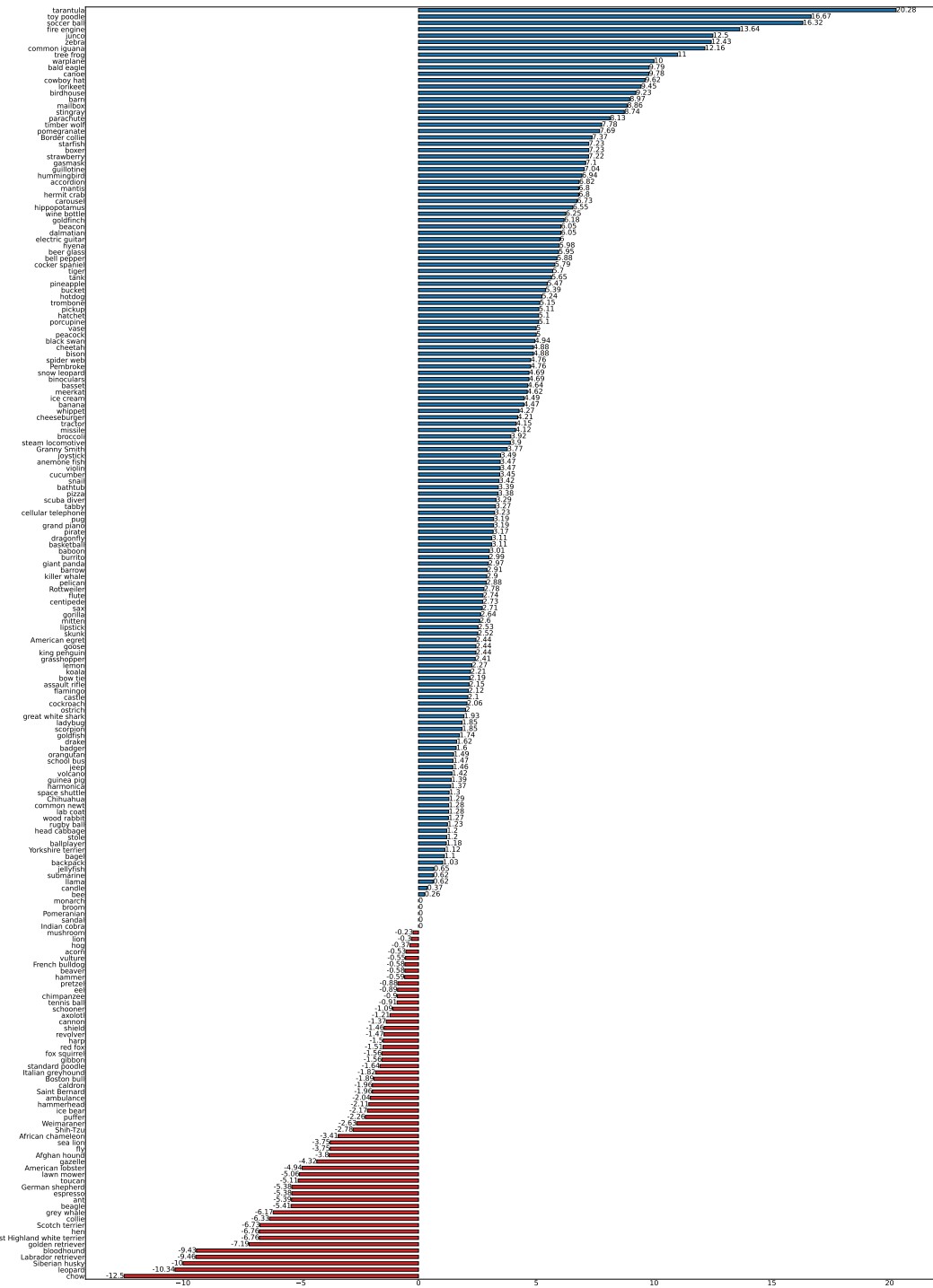

Figure 7: The effect of the finetune procedure on each class for ImageNet-R (there are some classes with zero effect).

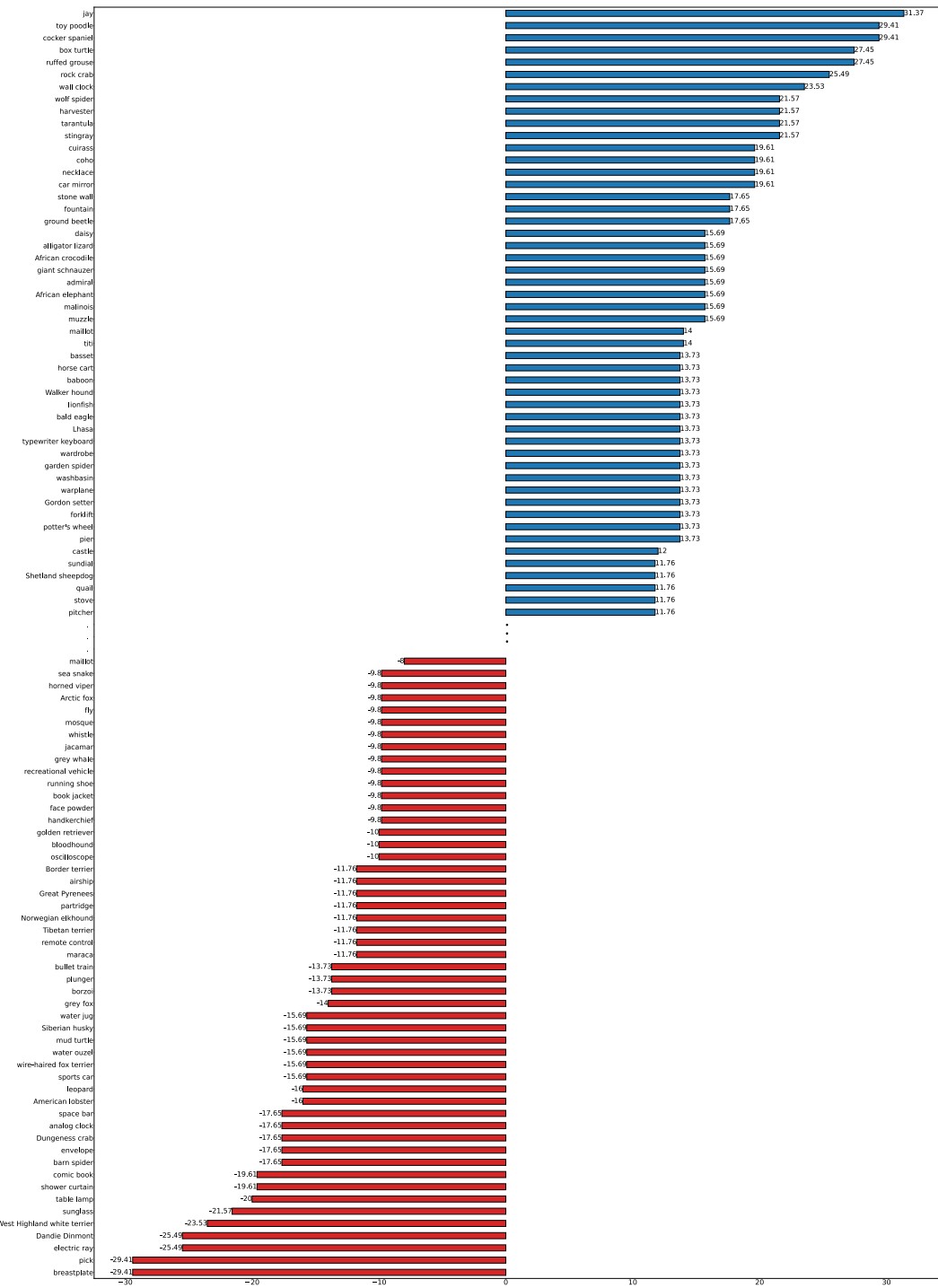

Figure 8: The 50 classes which most benefited and the 50 classes that were most harmed by the finetune procedure for ImageNet-sketch.

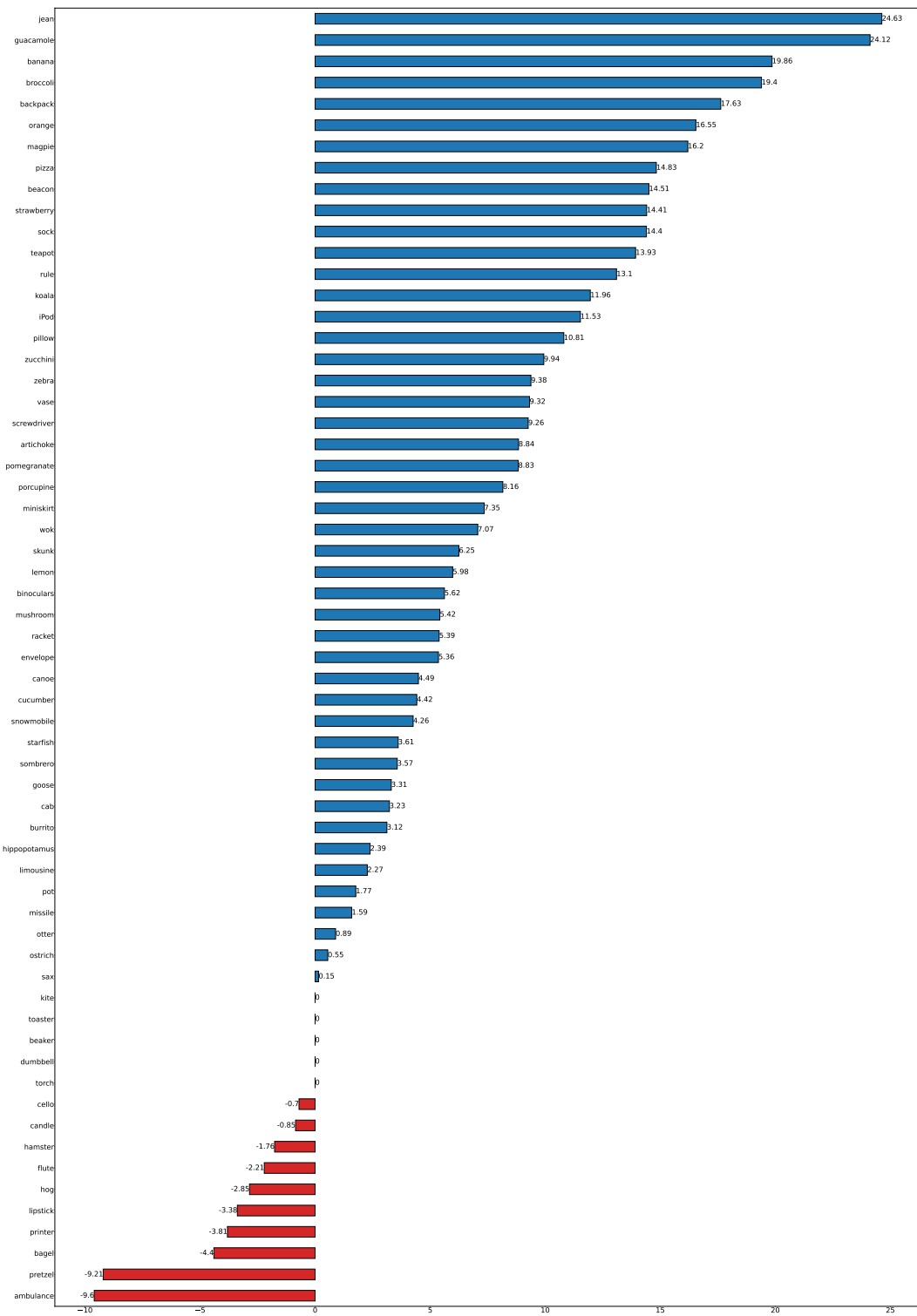

Figure 9: The effect of the finetune procedure on each class for SI-rotation (there are some classes with zero effect).

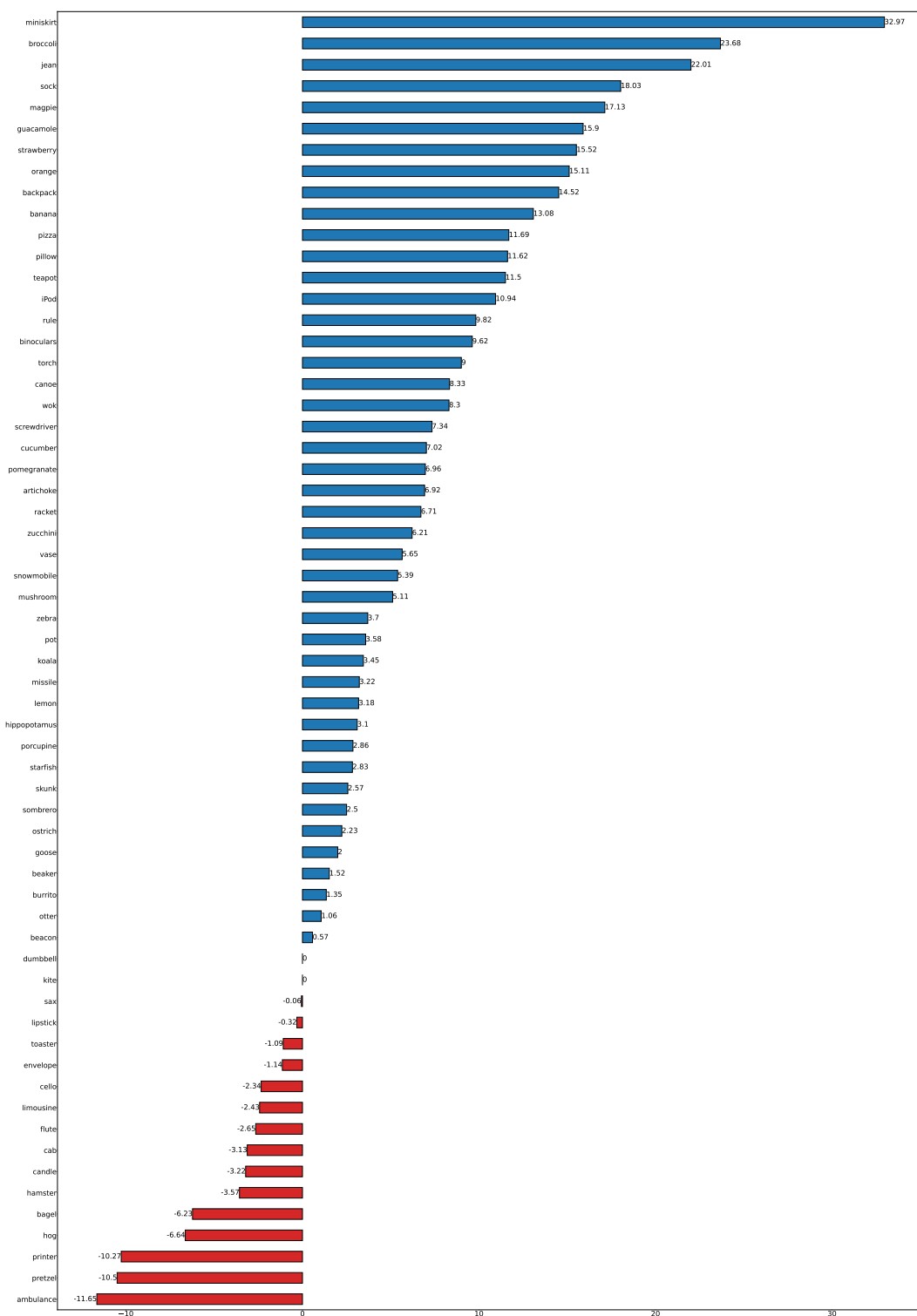

Figure 10: The effect of the finetune procedure on each class for SI-size (there are some classes with zero effect).

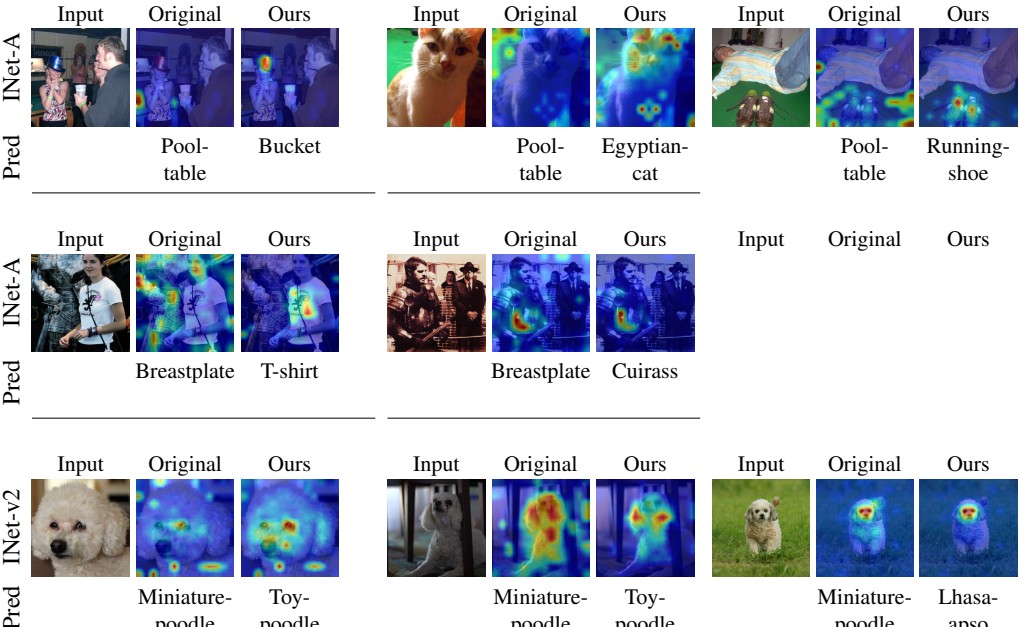

Figure 11: Examples of the 2 classes most harmed by our method for a dataset that is out of distribution (INet-A), and a dataset that has a similar distribution to ImageNet (INet-v2). For each dataset, we show the first three examples where our method modified a correct prediction and made it wrong (by showing the first three samples, we demonstrate that these samples are typical and not cherry-picked). As can be seen, in most cases the heatmaps are improved by our method, and the wrong prediction has a rationale (see text for examples). For INet-A there are only 2 mistakes in the Breastplate class (out of a total of 11 examples in the dataset), therefore the corresponding row (second row) only contains 2 examples.

## L    Comparison to self-supervised Vision Transformers

Our unsupervised finetuning is based on the DINO self-supervised Vision Transformer [4] (see Sec. 4). DINO's attention maps have demonstrated the ability to perform semantic segmentation without being trained with that objective. This property partially corresponds to the goal of our finetuning process (see Sec. 3).

In this section, we perform a comparison between the robustness granted by our method and the improved robustness by DINO. We compare two versions of DINO- the first, a self-supervised DINO with a linear head trained on ImageNet, and the second, a DINO network that was finetuned on ImageNet. For the linear probing version, the DINO backbone was frozen and a linear head was trained over the features by the frozen network, while for the finetuned version, the entire network was finetuned on ImageNet in a supervised manner.

Since DINO's code and finetuning are based on DeiT [24], we present a comparison between the original DeiT model (trained on ImageNet), our finetuned version of DeiT, and DINO. Additionally, we apply our method to the DINO version that was finetuned on ImageNet, since this version finetuned the entire network based on the supervised classification task, which may cause changes to DINO's salient behavior. In the latter case, we apply the same hyperparameters as we did for DeiT, i.e. a learning rate of $8e - 7$ (all other hyperparameters are identical for all models), for a fair comparison.

We note that throughout this section, we use unsupervised finetuning results of our method. This way, our method is applied in a way that is congruent with self-supervised learning.

Table 10: Comparison to self-supervised robustness performance by DINO [4] (Top-1 accuracy). The results are compared to DeiT [24] since DINO's code and finetuning are built on it. DINO linear stands for DINO with linear probing, and DINO finetuned stands for DINO with finetuning on ImageNet. For DINO finetuned we apply our method since the entire network was finetuned on a supervised classification task. All DINO models were taken from the official implementation.

| Model | Method | INet val | INet-A | INet-R | Sketch | INet-v2 | ObjNet | SI-loc. | SI-rot. | SI-size |
|---|---|---|---|---|---|---|---|---|---|---|
| DeiT-B | Original | 80.8 | 12.9 | 30.9 | 31.2 | 69.7 | 31.4 | 34.5 | 39.3 | 54.6 |
| | Ours | 80.5 | 18.3 | 32.8 | 31.5 | 69.3 | 36.3 | **37.8** | **44.2** | **59.3** |
| DINO-B | linear | 78.0 | 7.5 | 25.9 | 25.4 | 66.7 | 26.4 | 34.8 | 26.2 | 51.3 |
| DINO-B | finetuned | **82.6** | 16.4 | 34.9 | **35.4** | **72.0** | 34.9 | 35.4 | 40.0 | 55.3 |
| | Ours | 82.2 | **19.1** | **35.9** | **35.4** | 71.3 | **37.4** | 36.7 | 43.7 | 58.1 |

Table 11: Comparison to self-supervised robustness performance by DINO [4] (Top-5 accuracy). The results are compared to DeiT [24] since DINO's code and finetuning are built on it. DINO linear stands for DINO with linear probing, and DINO finetuned stands for DINO with finetuning on ImageNet. For DINO finetuned we apply our method since the entire network was finetuned on a supervised classification task. All DINO models were taken from the official implementation.

| Model | Method | INet val | INet-A | INet-R | Sketch | INet-v2 | ObjNet | SI-loc. | SI-rot. | SI-size |
|---|---|---|---|---|---|---|---|---|---|---|
| DeiT-B | Original | 94.2 | 31.0 | 44.2 | 48.6 | 86.8 | 48.5 | 54.6 | 56.3 | 73.4 |
| | Ours | 95.0 | 40.9 | 47.5 | 49.9 | 88.5 | 56.6 | **58.1** | 62.7 | **79.0** |
| DINO-B | linear | 93.8 | 23.5 | 38.1 | 42.4 | 86.7 | 47.0 | 53.9 | 46.2 | 73.4 |
| DINO-B | finetuned | **96.0** | 38.8 | 50.1 | 54.6 | **89.7** | 54.7 | 56.6 | 59.4 | 76.2 |
| | Ours | 95.7 | **42.6** | **51.7** | **54.7** | **89.7** | **58.0** | 57.7 | **63.1** | 78.7 |

Tab. 10, 11 present the top-1, top-5 results of our experiment, respectively. As can be seen, the linear probing version of DINO performs worst, below the original DeiT-B model. However, the finetuned version of DINO significantly improves DeiT accuracy and robustness. It should be noted that even when comparing DINO's finetuned version with our version of DeiT, our model outperforms DINO in 5 out of the 7 robustness datasets that are not from the ImageNet distribution (INet-A, ObjNet, SI-loc., SI-rot., SI-size), while the two others (INet-R, INet-Sketch) are datasets that contain sketches, cartoons, and art, for which our method is less effective (due to the absence

of background information). Additionally, applied to the finetuned version of DINO, our method improves robustness across all non-ImageNet datasets. Evidently, the finetuning of DINO had a negative effect on robustness for non-ImageNet datasets, which our method corrects.

These results further establish our method's significance. Even when a self-supervised robust network is used, the supervised finetuning process could cause changes for the worse in the salient behavior of the method, which are rectified by our method.

# M   Evaluation on non-ImageNet classes

In addition to the evaluation on ImageNet distribution shifts, we test the improvement of robustness on datasets that contain classes that do not appear on ImageNet.

We employ a $k$-NN test where we remove the classification head from the Vision Transformer, and use the [CLS] token as a global representation of the input. We consider three datasets for this experiment. First, we consider the iNat2021 mini dataset [14] which is a natural visual classification dataset, with $10k$ different classes. Secondly, we employ a medical dataset containing X-ray images of lungs to detect Pneumonia (binary classification) [15]. Finally, we consider the CIFAR-100 [16] dataset, which is an object categorization dataset containing 20 classes.

Fig. 12, 13, 14 present results for these three datasets. We employ both $k = 1$ and $k = 10$ neighbors, in order to demonstrate the effectiveness of our method in multiple settings ($k = 10$ is often better across all transformer models). As can be seen, our method improves the $k$-NN accuracy across different settings ($k = 1, k = 10$) and transformers, for all datasets, with the only exception being ViT AugReg small with $k = 1$ on the X-ray Pneumonia detection dataset. For challenging datasets where the models achieve under $50\%$ such as the iNat dataset, which contains $10k$ classes, our method improves over the original models by a very significant margin ($+5.37\%$ for $k = 1$, $+5.28\%$ for $k = 10$, averaged across all 7 models).

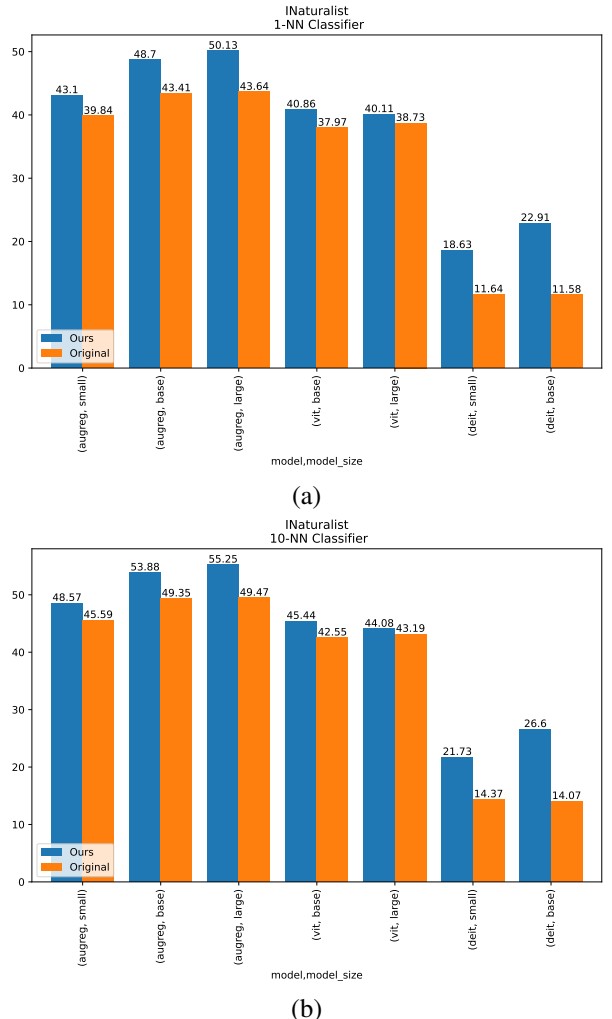

Figure 12: $k$-NN evaluation of performance on the iNat2021 mini dataset [14] before and after applying our method. (a) $k = 1$, (b) $k = 10$. Our method improves all models significantly.

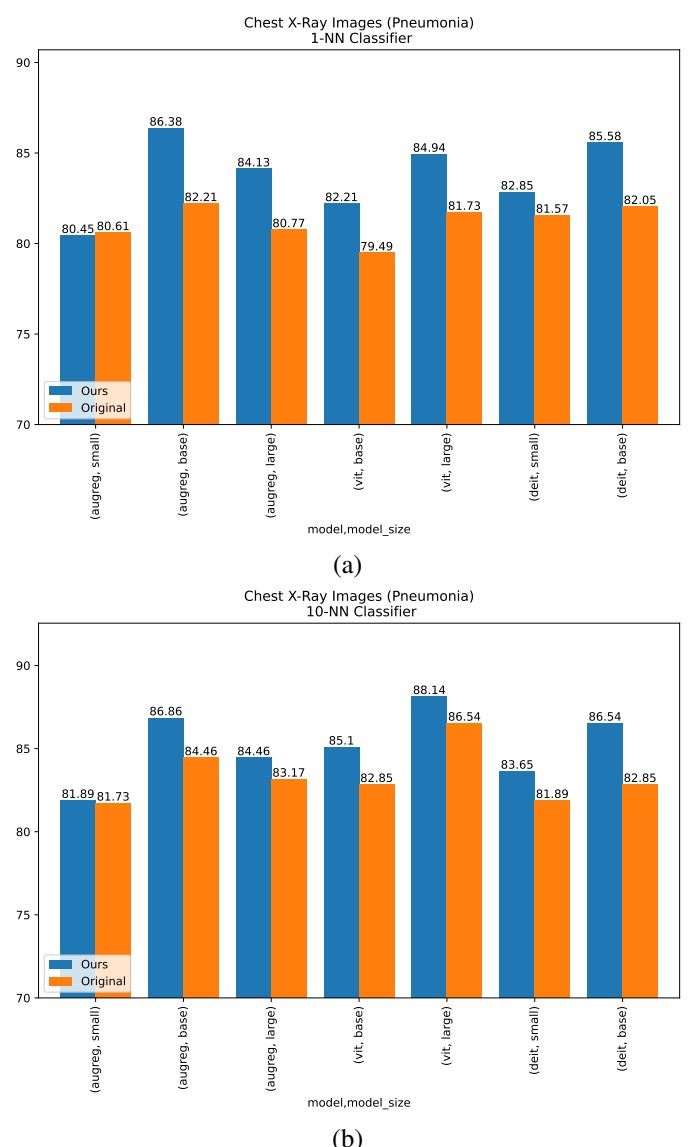

Figure 13: $k$-NN evaluation of performance on the X-ray Pneumonia dataset [15] before and after applying our method. (a) $k = 1$, (b) $k = 10$. Our method improves all models, other than the small variant of AugReg with $k = 1$. Note that the range of the y-axis does not start at zero.

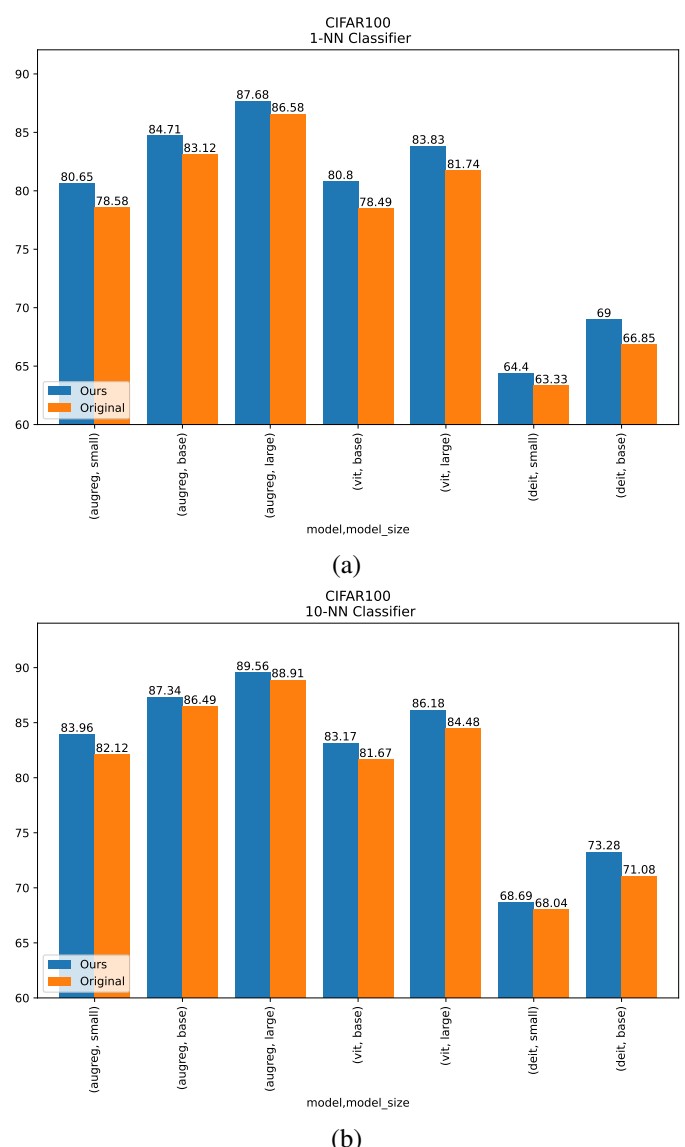

(a)

(b)

Figure 14: $k$-NN evaluation of performance on the CIFAR-100 dataset [16] before and after applying our method. (a) $k = 1$, (b) $k = 10$. Our method improves all models across all settings. Note that the range of the y-axis does not start at zero.

# N   Evaluating explanations

Our method applies loss terms directly to the explanations produced by GAE [7]. Under the assumption that GAE results in maps that faithfully reflect the model's reasoning, the result of our finetuning directly modifies the inner workings of the model such that the predictions by the model are not based on spurious correlations. In this section, we test the faithfulness of GAE before and after applying our method. It has already been shown in previous works [7, 17] that GAE is faithful with regard to the models before our manipulation. By examining the same metrics after applying our method, we ensure that the changes to the GAE maps indeed indicate a change in the behavior of the models, i.e. assuming that GAE is still faithful **after** employing our method, then indeed the positive changes to the GAE maps reflect a positive change in the salient behavior of the model.

To test faithfulness, we employ positive and negative perturbation tests following [8] on a random subset of $5,000$ images from the ImageNet validation set on the base models, i.e. ViT-B, DeiT-B, and AugReg-B. The positive and negative perturbation tests follow a two-stage setting. First, a pre-trained network is used for extracting visualizations for the validation set of ImageNet. Second, we gradually mask out the pixels of the input image and measure the mean top-1 accuracy of the network. In positive perturbation, pixels are masked from the highest relevance to the lowest, while in the negative version, from lowest to highest. In positive perturbation, one expects to see a steep decrease in performance, which indicates that the masked pixels are important to the classification score. In negative perturbation, a good explanation would maintain the accuracy of the model, while removing pixels that are not related to the class. In both cases, we measure the area-under-the-curve (AUC), for erasing between $10\% - 90\%$ of the pixels.

As can be seen from Tab. 12, the results of the perturbation tests indicate that GAE is still faithful after the finetuning we apply, and in some cases even achieves better scores than on the unchanged ViT models. This indicates that indeed the manipulation of GAE maps is translated to a difference in the inner workings of the models.

| Model | Positive perturbation | | Negative perturbation | |
|-------|-------------|-------|-------------|-------|
|       | Original    | Ours  | Original    | Ours  |
| ViT-B  | 15.91 | 14.95 | 51.43 | 53.54 |
| DeiT-B | 20.16 | 18.23 | 60.28 | 59.35 |
| AR-B   | 21.59 | 19.42 | 57.82 | 58.36 |

Table 12: Positive (lower is better) and negative (higher is better) perturbation tests for the base models (ViT-B, DeiT-B, and AugReg-B) before and after applying our method (AUC in percents).