# OpenReview forum: "Optimizing Relevance Maps of Vision Transformers Improves Robustness"
_NeurIPS.cc/2022/Conference — NeurIPS 2022 Accept_

### Official Review · Reviewer_FGu4 · 2022-06-27

**Rating:** 7
**Confidence:** 4
**Soundness:** 4 excellent
**Presentation:** 4 excellent
**Contribution:** 3 good

**Summary:**

When models are trained without extra guidance, they often extract spurious features that are causally irrelevant to the target task. The paper proposes to regularise "where the model looks at" with human-annotated segmentation masks (or with automatically-generated foreground masks). More precisely, the proposed method regularises Vision Transformers (ViT) by fine-tuning a normally pre-trained model with the objective of encouraging foreground regions to have greater "relevance map" [8] values. By aligning the model's attention with the true foreground, the proposed method improves the models' general robustness, measured in terms of ImageNet-A/R/V2/Sketch/ObjNet and SI-Score.

**Questions:**

See weaknesses above.

**Limitations:**

They are okay.

**Strengths And Weaknesses:**

## Strengths

The greatest strength of the paper is that it seems to solve an important problem in a relatively straightforward and intuitive fashion. Some might say it is unrealistic and costly to use human-annotated foreground masks. I strongly disagree with that viewpoint. The problem of spurious correlation and attention misalignment is not solvable without extra human guidance; the dataset itself does not contain sufficient guidance as to which feature a model must utilise to solve the problem. If it did, you won't see the consistent issue with spurious correlations in various models and datasets. Nonetheless, many researchers shun away from using extra annotations (especially if they are expensive like segmentations) and data points, for the fear that they are seen as impractical. I do not think so. I believe using such extra annotations can in fact be much cheaper in practice than providing the needed human guidance through complex loss and regularisation terms and hyperparameters (which require expensive model re-training to run HP search for every dataset and architecture).

The method does seem to improve the robustness across the board (Tab 1 and 2). There do exist some inconsistencies here and there, but they are not major and are expected for such large-scale evaluations. The internal mechanisms are also verified quite well through ablative studies (Tab 3 and 4). The extra results in Appendix are also quite impressive. Hyperparameter selection is described well in Section D. The proposed method has used the same set of hyperparameters across experiments, while the baselines use hyperparameters tuned for each setup. The sensitivity tests in Section G provide a nice insight that the method already improves the robustness quite a bit with only 100 additional segmentation masks (100 classes x 1 sample/class).

## Weaknesses

The main weakness is the choice of the explanation method - relevance map. This reduces the intuitiveness of the method quite a bit in my opinion. Based on my reading of Section B (yes, please put this important information in the main text!), the explanation is generated by
- computing gradient of ViT output w.r.t. attention map
- pointwise multiplying the gradient map above with attention values
- taking pointwise ReLU and head-wise averaging.
This is complicated. It is not intuitive as to "what effect will regularising such a complicated output of a ViT have on its inner mechanisms". For example, it would have been much easier to grasp the effect of directly regularising the attention map with the ground-truth foreground masks. You then understand that the ViT will be performing attention-weighted pooling across tokens where the weights are better aligned with the actual foreground features.

It would be great if the authors could explain in words or mathematical language how the regularisation of the relevance map would affect the model in question.

About hyperparameter tuning - could the authors explain for which objective metric the parameter tuning is executed? I have probably missed it. I'm worried if the tuning is performed with respect to the robustness measures. If so, the robustness information has leaked to the HP tuning and the good performances here will less likely generalise to other data and models.

Other minor comments:
- What is the authors' opinion on applying the technique on CNNs? It is in principle possible to generate the relevance maps for them. Maybe CAM-like heatmaps are more suitable there though.
- What stops us from performing the technique on training from scratch? Is it purely a computational limitation (single V100) ? It would be nicer to assure an improvement in robustness for the scratch-training scenario. That will also render the "classification" loss (Eq 4) unnecessary and reduce complexity.

## Conclusion

I weigh the strengths far more than the weaknesses. This is a nice paper that addresses an important problem with an intuitive approach. I hope the authors answer the remaining questions (and make minor revisions) to juice out the last bits of possible improvements.

---

> ### Author Response · Authors · 2022-07-29
> **Authors response (part i out of ii)**
>
> We thank reviewer FGu4 for the very positive and comprehensive comments. We appreciate the reviewer’s in-depth analysis of our work and attention to detail, including information and experiments provided in the appendices.
>
> Please note that all references to the text refer to the revised version of the paper
>
>  __Re. intuition for the explainability method__
>
> We thank the reviewer for bringing this point to our attention. Following the review, we have revised Appendix B to include further explanations and intuitions.
>
> Due to the 9-page limit, we were unable to move details from Appendix B to the main paper in the revised version of the paper. Section 3 of the paper will be edited with more details from Appendix B for the camera ready version, which will have an extra page.
>
> In a nutshell, using pure attention weights as an explanation raises two issues:
> 1. Each attention layer contains several attention heads. Previous works such as [Voita et al. Analyzing Multi-Head Self-Attention: Specialized Heads Do the Heavy Lifting, the Rest Can Be Pruned. ACL, 2019] demonstrates that different attention heads have different purposes, and not all heads contribute equally to the model's prediction.
> Therefore, performing aggregation over attention heads in each attention layer is not trivial, and simple averaging will account for irrelevant heads.
> 2. The Transformer architecture is built on several self-attention layers; each further contextualizes the input data and exchanges information between tokens. As such, it is unclear if the last attention layer tokens still represent the original input or a mixture of the input with context added by the previous layers.
>
> To mitigate both issues, GAE proposes to:
> 1. Use gradients as weights for the different attention heads. Highly important heads will receive a positive weight, while the unimportant ones will receive a negative or very small weight. The ReLU operation eliminates the negative contributions to avoid considering irrelevant heads. As a result, the head averaging is based on a weighting that considers the relevance of each head.
> 2. The integration of different layers is done using matrix multiplication of the relevances per layer.
>
> We chose to employ GAE specifically since previous works on the faithfulness of Transformer explainability [Liu et al. Rethinking Attention-Model Explainability through Faithfulness Violation Test. ICML 2022] found it to be the most faithful among all tested methods (including methods based on raw attention values). While we agree that attention has some relation to explanations, we believe that considering just raw attention is simplistic and, therefore, sub-optimal for the task of correcting the salient behavior of Transformers.
>
> To further substantiate our reasoning, we added two new ablations to our ablation study (Tab. 12 in Appendix I). The first replaces GAE with raw attention weights, and the second replaces it with attention Rollout [Abanar et al. Quantifying attention flow in transformers. ACL, 2020], which combines the attention maps in all layers linearly using matrix multiplication.
>
>
>
> The main conclusions from the experiment are summarized as follows:
> 1. For all datasets containing shifted distributions (i.e., all datasets except for INet val, INet-v2), our method with GAE outperforms the other variants.
> 2. For ViT-B, the use of raw attention (labeled “Attention instead of [9]”) harms some datasets significantly (e.g., INet-A, SI-rot.), while for DeiT-B, the use of Rollout (labeled “Rollout instead of [9]”) harms some datasets significantly (e.g., ObjNet, SI-loc.). This indicates that both variants are inconsistent and cannot be used as reliable explanations.
>
> Due to the current 9-page limit, the results are enclosed in Appendix I and will be moved to the main text for the camera-ready version, which will have an extra page.
>
> We hope this answers the reviewer’s concern but would be happy to further clarify otherwise.
>
> ---
>
> __Re. application for CNNs__
>
> We completely agree that this technique may be useful for CNN-based classifiers as well. We also concur that CAM-based explanations such as Grad-CAM [Selvaraju et al. Grad-CAM: Visual Explanations from Deep Networks via Gradient-based Localization. ICCV, 2017] can be applied in such a case. We opted to focus on ViTs as they are rapidly becoming the default model of choice for classification tasks.
>
> ---
>
> __Re. training from scratch__
>
> Indeed, limited resources played a large role in the selection to perform fine-tuning over training from scratch. While we believe that training with this regularization from scratch may produce an even higher increase in robustness, we note that there’s an advantage in a fine-tuning-based method as it can be applied to any model with relatively modest resources, and it produces results quickly.

---

> > ### Author Response · Authors · 2022-07-29
> > **Authors response (part ii out of ii)**
> >
> > __Re. hyperparameter tuning and robustness on other datasets__
> >
> > To expand over L. 170-173, the following process was followed due to resource limitations:
> > 1. The batch size of $8$ was the maximal size applicable for the amount of computing we have.
> > 2. All the hyperparameters (besides the learning rate, which was determined per model as described in L. 170-173) were determined by a grid search only on ViT-B and then applied to the other models without additional hyperparameter tuning.
> > 3. First, we performed a grid search between $[0, 1]$ (with jumps of $0.1$) on pairs of $\lambda_{\text{relevance}}$, $\lambda_{\text{classification}}$ (without using the foreground loss, i.e. only with $\mathcal{L}\_{\text{bg}}$). Our rule of thumb was to use the highest $\lambda_{\text{relevance}}$ and the lowest $\lambda_{\text{classification}}$ possible such that the validation set accuracy did not decrease by more than $2$% (similarly to the description in L. 170-173).
> > 4. Finally, we grid searched between $[0, 1]$ (with jumps of $0.1$) on pairs of $\lambda_{\text{bg}}$, $\lambda_{\text{fg}}$ with the same objective. $\lambda_{\text{fg}}$ could not be increased beyond $0.3$ without harming the validation accuracy by more than $2$%, and since $\lambda_{\text{bg}}=1$ produced visual results that still contained a lot of relevance on the background, we increased it to $\lambda_{\text{bg}}=2$, where the accuracy was not harmed significantly, but the visualizations of the saliency maps for samples from the validation set improved.
> >
> > The main objective of the grid search was to find hyperparameters such that the saliency maps of the validation set samples are improved while maintaining a similar accuracy to the original ViT-B model.
> >
> > We note that due to the large variety and the size of some of the datasets we used, it is impractical to perform a grid search based on the robust results of the datasets evaluated in our paper.
> >
> > Additionally, following the reviews and demonstrating our method’s improvement on completely unrelated datasets, we added a $k$-NN experiment in Appendix L of the revised paper, where we use datasets with classes that do not appear on ImageNet-1k.
> >
> > We experiment with 3 such datasets. First, the iNat 2021 mini dataset [Van Horn et al. Benchmarking Representation Learning for Natural World Image Collections. CVPR, 2021] tests the improvement of natural world image classification. Secondly, the Pneumonia detection dataset of X-ray images [Kermany et al. Identifying Medical Diagnoses and Treatable Diseases by Image-Based Deep Learning, 2018] is used to test the improvement of our method for medical data. Finally, the CIFAR-100 dataset is used as an additional classification dataset [Krizhevsky et al. Learning multiple layers of features from tiny images. 2009].
> >
> > The X-rays benchmark was selected simply since it is the first one that comes up on Google for the search “x-rays dataset deep learning”.
> >
> > We compare the baseline Transformers to the ones finetuned by our method on the TokenCut data, i.e., without any ground truth supervision in the form of manually extracted segmentation masks.
> >
> > The main conclusions from the experiment are summarized as follows:
> > 1. Our method improves the $k$-NN accuracy across different settings ($k=1, k=10$) and across the Transformer models for all datasets, with the only exception being ViT AugReg small with $k=1$ on the Pneumonia detection dataset.
> > 2. For challenging datasets where the models achieve under $50$% such as the iNat dataset (which contains $10k$ classes), our method improves over the original models by a very significant margin ($+5.37$% for $k=1$, $+5.28$% for $k=10$, averaged across all 7 models).
> >
> > Please refer to Appendix L for the full results.

---

> > > ### Comment · Reviewer_FGu4 · 2022-08-07
> > > **Thank you for the response - retaining my original score**
> > >
> > > Thank you for the detailed response.
> > >
> > > There were two major remaining questions that I have brought up:
> > > 1. How does the regularisation of GAE improve the actual recognition mechanism of the model?
> > > 2. How is the HP tuning performed? What was the objective for the HP tuning? If the objective is the robustness objective, then the HP tuning is leaking the final objective on top of the allowed resources.
> > >
> > > I do not think the authors have fully answered the questions.
> > >
> > > 1. The authors explain how and why **other** attribution/explanation methods are **not** working. While this is relevant and informative, they would have answered the question more directly if they had also explained the mechanism behind their method in a greater detail. It is not very intuitive how regularising such a derivative score map of the Transformer architecture leads to the change in the actual mechanism of the original model. The model could simply learn to overfit its GAE score map to the segmentation GT, while actually not changing the actual inner workings of the model. What prevents this from happening? Indeed, the proposed regularisation of GAE is already producing improved robustness performances - which in itself tells us that something must be happening inside. But it would have been more illustrative if the authors could further *show* the mechanism more directly.
> > >
> > > 2. The main concern behind the question is that one could introduce information leakage during HP tuning. For example, by tuning HP wrt  the "the saliency maps of the validation set samples", one is introducing more number of segmentation GTs than claimed in the paper. In principle, the method is using 3 segmentation GTs per class during training + X segmentation GTs per class in the validation set during HP tuning. From the response, I can infer that the HP tuning procedure leaks certain amount of additional segmentation GTs. This is fine, but I believe the HP tuning procedure should be clearly stated and shared with the readers.
> > >
> > > Given all that, I still believe the paper is strong and should be accepted. I'm retaining my score.

---

> > > > ### Author Response · Authors · 2022-08-08
> > > > **Thank you for this discussion! (addressing question 1 out of 2)**
> > > >
> > > > We are grateful to the reviewer for the detailed feedback and for taking the time to engage in discussion. We apologize for the slight delay in our response which is due to the runtime of the experiments we present in our answers below.
> > > >
> > > > To address the remaining questions:
> > > >
> > > > __Re. using GAE to improve recognition__
> > > >
> > > > > “The model could simply learn to overfit its GAE score map to the segmentation GT, while actually not changing the actual inner workings of the model. What prevents this from happening?”
> > > >
> > > > To our understanding, and please correct us otherwise, the reviewer asks whether it may be possible that the method improves the loss by changing the explainability map without affecting the classifier itself. While the reviewer notes that the robustness scores improve (and so the classifier has changed), it is still not clear whether applying a loss on the explainability map would not move to detach it from the underlying classifier.
> > > >
> > > > As we mentioned in our original response, works on evaluating faithfulness of Transformer explainability such as [35] found GAE to be most faithful, therefore the GAE mechanism produces maps that are highly correlated with the explanation of the model. Assuming this link remains after applying our method, optimizing the maps produced by GAE directly impacts the reasoning by the model.
> > > >
> > > > We, therefore, conduct tests to demonstrate that GAE indeed constitutes a faithful explanation after applying our method. By testing the faithfulness after applying the method, we show that our fine-tuning modifies the underlying explanation of the prediction, which is mirrored by GAE, thus the improved explanations presented in the paper indeed reflect the reasoning by the model.
> > > >
> > > > We conduct positive and negative perturbation tests for the base models (ViT-B, DeiT-B, and AugReg-B) before and after applying our method and present the area under the curve for both tests. These tests follow a two-stage setting. First, the model is used for extracting visualizations for the validation set of ImageNet. Second, we gradually mask out the pixels of the input image and measure the mean top-1 accuracy of the network. In positive perturbation, pixels are masked from the highest relevance to the lowest, while in the negative version, from lowest to highest. In positive perturbation, one expects to see a steep decrease in performance, which indicates that the masked pixels are important to the classification score. In negative perturbation, a good explanation would maintain the accuracy of the model, while removing pixels that are not related to the class. In both cases, we measure the area-under-the-curve (AUC), for erasing between $10$%-$90$% of the pixels.
> > > >
> > > > The results of the perturbation tests are presented in Appendix N of the latest revision and indicate that GAE is still faithful after the finetuning we apply, and in some cases even achieves better scores than on the unchanged ViT models. Thus it is unlikely that optimizing our loss on the GAE relevance map detaches it from the fine-tuned model.
> > > >
> > > > > “How does the regularisation of GAE improve the actual recognition mechanism of the model? … It is not very intuitive how regularising such a derivative score map of the Transformer architecture leads to the change in the actual mechanism of the original model.”
> > > >
> > > > Assuming that the question is specifically about GAE and the intuition behind it, we would like to note that we have revised Appendix B of the paper with an in-depth explanation and intuition for the GAE method. We would be happy to discuss this further and answer any questions the reviewer may have.
> > > >
> > > > Additionally, we kindly note that GAE is a supportive method we use in order to achieve our goal, and it can be replaced with other methods that extract faithful explanations for the model’s prediction. While GAE seems to be the most accurate explainability method for Vision Transformers, other explainability methods can be used to improve the salient behavior of models. This is demonstrated by the ablations added in Tab. 12 in Appendix I, which show that while other explainability methods fall short in comparison to GAE, our method is able to improve robustness even when using less reliable methods.

---

> > > > > ### Author Response · Authors · 2022-08-08
> > > > > **(addressing question 2 out of 2)**
> > > > >
> > > > > __Re. hyperparameter tuning__
> > > > >
> > > > > We thank the reviewer for the question, and agree that it is highly important to disclose all the details of the hyperparameter tuning process and to count the number of segmented images correctly.
> > > > >
> > > > > While the validation set was indeed used to tune the hyperparameters, please note that:
> > > > > 1. The validation set only contains 414 examples (L. 166-167), such that even if we consider these examples as additional segmentation supervision, the validation set adds less than a single example per class.
> > > > > 2. As we mentioned in our previous reply, the hyperparameter tuning was only conducted on ViT-B. The parameters were then applied without further modification to all other models.
> > > > > 3. The exact same hyperparameters were applied as is to the unsupervised tests (with TokenCut), further indicating the stability of our method.
> > > > > 4. It should be mentioned that our method is not sensitive to the specific selection of hyperparameters, and small changes to the selection maintain a similar improvement in robustness (L. 171-173).
> > > > > 5. Assuming that only 3 labeled segmentation maps are available per class, the hyperparameter tuning process could be performed by splitting the training set to a train and validation set, where the training set contains two segmentation maps per class and the validation set consists of a single segmentation map per class. This is based on the results in Appendix G that indicate that even 2 segmentation maps per class suffice to achieve a significant improvement in robustness, in addition to the fact that currently we use 414 validation samples, therefore such a split should produce very similar hyperparameters to the ones selected by our full method.

---

> > > > > > ### Comment · Reviewer_FGu4 · 2022-08-09
> > > > > > **Thanks!**
> > > > > >
> > > > > > Thank the authors for an extended discussion.
> > > > > >
> > > > > > 1. Whether supervising GAE faithfully changes the inner mechanisms of the Transformer
> > > > > >
> > > > > > After going through the references [8] on GAE and the ICML 2022 paper [35] evaluating multiple explanation methods on attention-based models, I'm convinced that GAE is indeed faithful. I'm also convinced that supervising GAE will improve the faithfulness of the underlying model, based on the shown results in the paper.
> > > > > >
> > > > > > After the discussion, I think the source of my dissatisfaction has to do with the relevance maps themselves and not the submission. I still do not understand the reasoning behind the pointwise multiplication between the activation and gradient maps and the pointwise positive clipping that follows. This often works well in practice, but that does not mean that the particular set of operations is very interpretable to me. Having said that, I understand that this is way beyond the scope of the submission and therefore this shall not affect my score for this paper.
> > > > > >
> > > > > > 2. Hyperparameter tuning
> > > > > >
> > > > > > Thanks a lot for clarifying the resources used for HP tuning and pointing to the parts that I have missed or failed to recall. I agree that the additional information leakage is not great and the method is robust against the choice of HP.
> > > > > >
> > > > > > Thanks again for the clarification.

---

### Official Review · Reviewer_y4aF · 2022-07-08

**Rating:** 6
**Confidence:** 3
**Soundness:** 3 good
**Presentation:** 2 fair
**Contribution:** 2 fair

**Summary:**

This paper presents a method for robustifying Transformers-based image classifiers against different image distributions, assuming that better attention improves the generalization performance. The main idea is to force the relevance map (or an aggregation of attention maps in Transformers) to focus on foreground objects. The method generates a relevance map with [8] and gives a manual or unsupervised segementation map as a preferred relevance map. The advantage of the method is experimentally demonstrated.

**Questions:**

1. I would like to see some discussions and explanations on Weaknesses 1-3.
1. For the unsupervised segmentation case, how many images are used for training with segmentation results?
2. I'm also curious about the performance of TokenCut over the datasets used in the paper.

**Limitations:**

I'm not very sure if this is a limitation, but the paper does not mention the distribution used for training the unsupervised segmentation model. At least segmentation may not work when the data distribution is completely different (e.g., X-ray images). I think this point is not discussed in the paper.

**Strengths And Weaknesses:**

## Strengths

1. The method is simple but actually improves the performance.
2. A good set of comparisons and ablation are presented.

## Weaknesses

1. According to Eqs. (6) and (7), a relevance map is based on gradients. I guess the classifier (or the model) is trained through relevance maps, and if this is the case, more details on how the model is trained with these gradients.
2. There are some methods that try to debias a model, like [Burns et al., "Women also Snowboard: Overcoming Bias in Captioning Models," ECCV 2018], which also uses manual segmentation for debiasing the model. I think the proposed method shares the basic ideas with such methods, and they may be easily adapted to classification tasks. I would like to see how the proposed method differs from these methods. Experimentally comparing some of these methods (for the manual segmentation variant) or at least providing discussion on them could be beneficial if they are comparable.
3. The paper could discuss the data distribution used for training TokenCut in the context of generalization of the method to severely different data distributions.

---

> ### Author Response · Authors · 2022-07-29
> **Authors response (part i out of ii)**
>
> We thank reviewer y4aF for the detailed feedback and the useful suggestions.
>
> Please note that all references to the text refer to the revised version of the paper.
>
> Addressing items 1–3 mentioned as weaknesses, and the additional questions:
>
> __Re: training with gradients (Eq. 6,7 in the original submission, Eq. 8,9 in the revised version)__
>
> Assuming we understood the question correctly, and please correct our understanding otherwise, the reviewer is asking how the gradients in Eq. 6,7 (Eq. 8,9 in the revised version) are used in the fine-tuning process described in the paper.
>
> These equations, which appear in Appendix B, describe how the explainability method of GAE  computes the relevancy maps. This computation relies, among other components, on the attention gradients.
>
> Our method employs the relevancy maps calculated in Eq. 6,7 (Eq. 8,9 in the revised version) to construct relevance-based losses. We kindly refer the reviewer to Section 3 of the main paper (Eq. 1,2) which specifies the two losses constructed using the relevance maps ($\mathcal{L}\_{\text{fg}}$, $\mathcal{L}\_{\text{bg}}$).
>
> To optimize these loss terms using SGD, a gradient is calculated on top of the relevance maps. Since the relevance maps involve gradients themselves, this means that second-order gradients are calculated during SGD in our fine-tuning. The losses we apply are calculated directly on the relevance maps, which are derivable as a combination of pure attention weights and attention gradients.
>
> Please see the revision of Appendix B of the paper for extended details on the calculation of the relevancy maps, and an intuitive explanation of the method.
>
> We would be happy to further clarify or answer any other questions the reviewer may have about our method.
>
> ---
>
> __Re: comparison to debiasing methods__
>
> We thank the reviewer for pointing us to the ECCV’18 work.
> The goal of the ECCV’18 work is to eliminate gender bias in a CNN-based image captioning system. The method is based on masking the person in the image (the person's segmentation map is provided). For the masked samples, the method reinforces the decision not to distinguish between a man and a woman through a loss term that is called the Appearance Confusion Loss.
>
> The two methods use segmentation maps, but there are key methodological differences beyond the completely different goals (classification vs. captioning, improving robustness vs. removing a single specific bias) and settings (Transformers vs. CNNs, fine-tuning vs. training from scratch). Most notably: our method optimizes relevancy maps directly, while the ECCV’18 work uses losses that are applied to the output distribution of the model.
>
> Exporting their idea from the task of eliminating gender bias in image captioning to image classification is not trivial.
> It is possible to mask the object using the segmentation map as a straightforward adaptation but concealing an object inherently differs from masking a person.
>
> By masking a person, gender is obscured (the silhouette of a woman and a man is indistinguishable). Masking an object, however, still reveals significant information about the object through its shape. For example, after masking a snake, it would still be clear that the class is not "table", "cat" or "dog".
>
> The second challenge is defining the confusion loss (e.g., the original loss involves confusion between men and women). In the case of classification, an alternative loss can require a uniform class distribution given the masked image. However, this will probably lead to a severe accuracy hit, since, as mentioned, the masked image should not receive uniform scores across different classes.
> In the snake example, we would not expect the model to output a uniform distribution, as this would assign "table" the same probability as "water snake" for example. We would still expect the distribution to be peaked with snake classes receiving a high probability while the other classes receive a probability close to 0. Therefore, this adaptation of the ECCV’18 method to classification is counter-intuitive and probably harmful.
>
> While it is not directly applicable in our setting, we appreciate the reviewer for bringing this work to our attention. Inspired by this discussion, we have added a section referring to debiasing methods similar to that of ECCV’18 (see Sec. 2 L. 82-86).
>
> Please let us know if this discussion does not address your concern regarding the ECCV’18 work.
>
> ---
>
> __Re. Unsupervised fine-tuning setting__
>
> The unsupervised fine-tuning setting is identical to the setting of the supervised version, i.e. we use 3 examples from 500 ImageNet-1k classes. The only difference is that in the unsupervised case, the segmentation maps for the fine-tuning examples are tagged using TokenCut, as opposed to manual human tagging in the supervised setting.

---

> > ### Author Response · Authors · 2022-07-29
> > **Authors response (part ii out of ii)**
> >
> > __Re: TokenCut training distribution and robustness__
> >
> > TokenCut employs a pre-trained DINO network [Caron et al.. Emerging properties in self-supervised vision transformers. ICCV, 2021], which is trained on the ImageNet-1k data without labels via self-distillation.
> >
> > Following the reviews, we have added Appendix K to the revised version of the paper, where we present comparisons between DINO and our unsupervised method. To maintain a fair and unbiased comparison, we benchmark DINO against DeiT and our fine-tuned version of it, since as mentioned in their paper, DINO training and fine-tuning are based on DeiT.
> >
> > We employ 2 variants of DINO in our comparison. The first is a linear probing version, where a linear classification head was trained on top of a frozen network, and the second is a fine-tuned version of DINO where the entire network was modified. The application of our method to DINO after the authors’ ImageNet fine-tuning process could be beneficial, since their fine-tuning is done in a supervised manner, and could impact the salient behavior of DINO for the worse (i.e. eliminate some of the robustness benefits that arise from self-supervision training).
> >
> > In both cases, the application of our method does not use supervision in the form of manually labeled segmentation masks. This way, our method is applied in a way that is congruent with self-supervised learning.
> >
> > The main conclusions are summarized as follows:
> > 1. The linear probing version of DINO is not able to measure up against the performance of even the original, unchanged DeiT model.
> > 2. The fine-tuned version of DINO significantly improves accuracy and robustness over the original, unchanged DeiT model.
> > 3. Even when comparing DINO's fine-tuned version with our version of DeiT, our model outperforms DINO in $5$ out of the $7$ robustness datasets that are not from the ImageNet distribution (INet-A, ObjNet, SI-loc., SI-rot., SI-size), while the two others (INet-R, INet-Sketch) are datasets that contain sketches, cartoons, and art, for which our method is less effective (due to the absence of background information).
> > 4. Our method improves robustness for the fine-tuned version of DINO, indicating that the supervised fine-tuning process could cause changes for the worse in the salient behavior of the method, which are rectified by our method.
> >
> > Please refer to appendix K of the revised version for additional details and the full results.
> >
> > ---
> >
> > __Re. TokenCut for entirely different distributions (e.g. X-rays)__
> >
> > While TokenCut was trained on ImageNet, the improved representations by our method have a significant positive influence even on data from distributions that are inherently different than the ImageNet-1k distribution.
> >
> > We demonstrate this point by adding a $k$-NN experiment in Appendix L of the revised paper, where completely different datasets are used. The Pneumonia detection in X-ray images dataset [Kermany et al. Identifying Medical Diagnoses and Treatable Diseases by Image-Based Deep Learning, 2018] is used to test the improvement by our method for medical data, the iNat 2021 mini dataset [Van Horn et al. Benchmarking Representation Learning for Natural World Image Collections. CVPR, 2021] tests the improvement on natural world image classification, and CIFAR-100 is an additional classification dataset [Krizhevsky et al. Learning multiple layers of features from tiny images. 2009].
> >
> > Note that the X-rays benchmark was selected based on the reviewer’s example and that this specific dataset is simply the first one that comes up on Google for the search “x-rays dataset deep learning”.
> >
> > We compare the baseline Transformers to the ones finetuned by our method on the TokenCut data, i.e., without any ground truth supervision in the form of manually extracted segmentation masks.
> >
> > The main conclusions from the experiment are summarized as follows:
> > 1. Our method improves the $k$-NN accuracy across different settings ($k=1, k=10$) and across the Transformer models for all datasets, with the only exception being ViT AugReg small with $k=1$ on the Pneumonia detection dataset.
> > 2. For challenging datasets where the models achieve under $50$% accuracy such as the iNat dataset (which contains $10k$ classes), our method improves over the original models by a very significant margin ($+5.37$% for $k=1$, $+5.28$% for $k=10$, averaged across all 7 models).
> >
> > Please refer to Appendix L for the full results.

---

> ### Author Response · Authors · 2022-08-07
> **We would be happy to address any follow-up questions**
>
> Thank you again for your detailed feedback and useful ideas.
>
> We would respectfully like to follow up to see if our response addresses your concerns. We would appreciate the opportunity to discuss our work further if the response has not already addressed all concerns.

---

> > ### Comment · Reviewer_y4aF · 2022-08-08
> > **Response to Paper2489 Authors**
> >
> > I appreciate the efforts made by the authors to address my concerns and am sorry for the late participation in the discussion.
> >
> > **Re: training with gradients (Eq. 6,7 in the original submission, Eq. 8,9 in the revised version)**
> >
> > For this point, I actually didn’t know that second-order derivatives can be computed by neural network frameworks, as they require the computational graph even for the gradient computation. But according to the authors’ response, this is doable, and so my concern is addressed.
> >
> > **Re: comparison to debiasing methods**
> >
> > I think both this paper and ECCV’18 paper share the idea of forcing the relevance map (or attention) to focus on a relevant region (for this paper, it’s foreground regions, and for ECCV’18 paper, it’s people regions). But as in the authors' response, the way it is optimized is very different, and I agree with the authors that it’s not straight-forward and not trivial to adopt the ECCV’18 method to classification tasks (for example, one may occasionally remove the foreground region by blocking its bounding box so that the shape doesn’t tell what it is and setting the ground-truth to a new label “none of them” or the uniform distribution, which seems not informative for training).
> >
> > My point here was that, at least for me, ECCV’18 looked to give something like contrastive supervision that inherently told where to see in the image, while directly optimizing relevance maps might still have a chance to superficially optimize them, leading to less generalizability. But again, the experimental results show its generalization performance, so I think adding some discussions on this family of work is sufficient.
> >
> > **Re. TokenCut for entirely different distributions (e.g. X-rays)**
> >
> > This conclusion is surprising to me. I appreciate the effort.
> >
> > For the other points, I think the authors' responses are satisfactory for me. I'm happy to increase my rating.

---

> > > ### Author Response · Authors · 2022-08-08
> > > **Thank you very much for participating in the discussion.**
> > >
> > > We appreciate the careful consideration of our response and acknowledge the need to discuss the ECCV'18 work and other work of the same family.
> > >
> > > We notice that while you wrote "I'm happy to increase my rating" the score has not been raised yet.

---

### Official Review · Reviewer_KJ1F · 2022-07-10

**Rating:** 7
**Confidence:** 4
**Soundness:** 3 good
**Presentation:** 3 good
**Contribution:** 3 good

**Summary:**

The paper focuses on improving the robustness of Vision Transformers by monitoring the relevancy map of models. Acted as a fine-tuning step, the proposed method contains three losses to suppress relevance on background regions, force the model to predict using foreground information, and learn from its own predictions. Experiments on several datasets show the effectiveness of the method.

**Questions:**

1. Provide a baseline experiment using DINO pretraining checkpoints.
2. What if the out-of-distribution datasets have some semantic-different classes, would the K-nearest neighbor testing on these novel classes still show better performance than the baseline? As mentioned in the paper, only parts of the classes used during fine-tuning improve the performance of the other classes. Therefore, the method should also work well in unseen classes?


**Ethics Review Area:**

["I don’t know"]

**Limitations:**

Yes, the authors have addressed the limitation. The potential negative societal impact is not mentioned, which is okay as the paper focuses on improving the robustness of general models.

**Strengths And Weaknesses:**

Strengths:
The motivations in the paper were well established. It is a relatively simple idea, but the main claims are validated by experimental results and visualizations.

Weakness:
1. The paper should consider some self-supervised learning (SSL) methods using ViTs as a baseline, as SSL methods are believed to work well on out-of-distribution than supervised methods. Besides, DINO [1] also shows the property to identify the foregrounds, which may, to a certain extent, remedies the shortcoming of relying on image backgrounds to classify.  Providing the performance of SSL ViTs will make the paper more convincing.
2. Adding the $L_{classification}$ on the fine-tuning phase, the method is limited to pre-trained classes. What if the out-of-distribution datasets have some semantic-different classes (iNaturalist or other natural datasets with different classes), would the K-nearest neighbor testing of these novel classes still show better performance?
3. Also, the organization of the paper is not ideal. For instance, in Section 2, the description of validation datasets and related works are mixed together.

---

> ### Author Response · Authors · 2022-07-29
> **Authors response**
>
> We thank reviewer KJ1F for the positive comments and useful suggestions. We believe the experiments suggested by the reviewer will have a significant positive impact on the quality of our work, and for that, we express our sincere gratitude.
>
> Please note that all references to the text refer to the revised version of the paper.
>
> __Re. baseline experiment using DINO__
>
> We concur that comparing to DINO is intriguing, especially since the unsupervised version of our method employs TokenCut, which is based on DINO’s attention maps.
>
> In Appendix K of the revised version of the paper, we present comparisons between DINO and our unsupervised method. To maintain a fair and unbiased comparison, we benchmark DINO against DeiT and our fine-tuned version of it, since as mentioned in their paper, DINO training and fine-tuning are based on DeiT.
>
> We employ 2 variants of DINO in our comparison. The first is a linear probing version, where a linear classification head was trained on top of a frozen network, and a fine-tuned version of DINO where the entire network was modified. The application of our method to DINO after the authors’ ImageNet fine-tuning process could be beneficial, since their fine-tuning is done in a supervised manner, and could impact the salient behavior of DINO for the worse (i.e. eliminate some of the robustness benefits that arise from self-supervision training).
>
> In both cases, the application of our method does not use supervision in the form of manually labeled segmentation masks. This way, our method is applied in a way that is congruent with self-supervised learning.
>
> The main conclusions are summarized as follows:
> 1. The linear probing version of DINO is not able to measure up against the performance of even the original, unchanged DeiT model.
> 2. The fine-tuned version of DINO significantly improves accuracy and robustness over the original, unchanged DeiT model.
> 3. Even when comparing DINO's fine-tuned version with our version of DeiT, our model outperforms DINO in $5$ out of the $7$ robustness datasets that are not from the ImageNet distribution (INet-A, ObjNet, SI-loc., SI-rot., SI-size), while the two others (INet-R, INet-Sketch) are datasets that contain sketches, cartoons, and art, for which our method is less effective (due to the absence of background information).
> 4. Our method improves robustness for the fine-tuned version of DINO, indicating that the supervised fine-tuning process could cause changes for the worse in the salient behavior of the method, which are rectified by our method.
>
> Please refer to Appendix K of the revised version for additional details and the full results.
>
> ---
>
>
> __Re. K-nearest neighbor testing on novel classes__
>
> We thank the reviewer for this suggestion and agree that a $k$-NN experiment on datasets with different classes is indeed an interesting way of testing the improvement of latent representations by our method.
>
> Appendix L of the revised paper presents $k$-NN results for 3 such datasets: iNat2021 mini [Van Horn et al. Benchmarking Representation Learning for Natural World Image Collections. CVPR, 2021]  (as suggested by the reviewer), the Pneumonia detection in X-ray images dataset [Kermany et al. Identifying Medical Diagnoses and Treatable Diseases by Image-Based Deep Learning. 2018], and CIFAR-100 [Krizhevsky et al. Learning multiple layers of features from tiny images. 2009].
>
> The X-rays benchmark was selected simply since it is the first one that comes up on Google for the search “x-rays dataset deep learning”.
>
> The main conclusions from the experiment are summarized as follows:
> 1. Our method improves the $k$-NN accuracy across different settings ($k=1, k=10$) and all Transformer models for all datasets, with the only exception being ViT AugReg small with $k=1$ on the Pneumonia detection dataset.
> 2. For challenging datasets where the models achieve under $50$% such as the iNat dataset (which contains $10k$ classes), our method improves over the original models by a very significant margin ($+5.37$% for $k=1$, $+5.28$% for $k=10$, averaged across all seven models).
>
> Please refer to Appendix L for the full results.
>
> ---
>
> __Re. paper organization in Sec. 2__
>
> We thank the reviewer for bringing this to our attention. We have edited Section 2 in the revised version to include titled paragraphs and changed the order and some of the content to allow for better readability.

---

> > ### Comment · Reviewer_KJ1F · 2022-08-07
> > **Thanks for  explanation and additional experiments**
> >
> > Thanks for your explanation and additional experiments.
> >
> > The authors provide an additional comparison with DINO and kNN results on novel classes, which strengthens the paper and addresses my main concerns. Thus, I increase my rating.

---

### Official Review · Reviewer_Tjbv · 2022-07-12

**Rating:** 6
**Confidence:** 4
**Soundness:** 3 good
**Presentation:** 2 fair
**Contribution:** 3 good

**Summary:**

This submission aims to improve the robustness of vision transformers (ViT) by leveraging interpretability methods during training. The proposed approach relies on a recent method for computing pixel-wise relevance maps for ViT models. The relevance map for a pre-trained ViT does not necessarily coincide with the area occupied by the foreground object. Accordingly, a pre-trained model is fine-tuned with cross-entropy loss and two regularisation terms aimed at improving the relevance map. These two terms respectively encourage positive and negative agreement of the relevance map with a foreground segmentation map. This method is evaluated on several ImageNet-adjacent robustness benchmarks, and results in either noticeable improvements (>1pts: ObjNet, INet-A, INet-R) or performance on par with the baseline (+/-1pt: INet, INet-v2, Sketch).



**Questions:**

I have a few questions about the experimental evaluation:

I would have liked to see a baseline which fine-tunes the model only using the classification loss. The reported results labelled "original" appear to refer to performance for the pre-trained model without fine-tuning. Is this correct? As such, we don't quite have a proper comparison between methods.

I am also very confused by the choice of hyperparameters for the competing methods (RRR, GradCAM) -- see Appendix. I don't understand why the weight for the classification loss was not kept static for all methods together with the learning rate. What happens if you conduct a grid search for the regularisation term weights by fixing the aforementioned hyperparameters. Could you also describe the "difficulties" you faced while choosing the appropriate hyperparam values?

While it is interesting that the method is effective given very little data, it should however be mentioned that (based on Fig. 8 in the appendix) the number of classes (500) and number of samples per class (3) were selected based on accuracy. Increasing the number of classes and/or samples beyond the chosen values can have a slightly negative impact, especially visible for the ImageNet-A curve.

This is a little surprising to me and I wonder why that is. Was the number of fine-tuning steps adapted to a change in the number of samples/classes? Is then the improvement a function of then number of training steps, or does the diversity of fine-tuning data not matter all? This is not immediately clear based on Fig. 8. It thus seems like a relatively minor adaptation of the network weights is enough to get better relevance maps and with it improved robustness. On which layers does the fine-tuning have the largest impact?

I also think that an opportunity was missed when it comes to examining the impact of different loss terms. I could not find an ablation study that considers different combinations of the regularisation terms that were considered (e.g. fg only, bg only).

What is the difference between AugReg and the corresponding vanilla ViT in terms of training?



**Limitations:**

Negative societal impact of the work is not addressed ([N/A] in the check list).

**Strengths And Weaknesses:**

This is a nice simple idea and it appears to improve robustness without much effort. If I understand correctly, it relies on fine-tuning a model for 50 epochs with just 1500 images per epoch (500 classes, 3 images each -- half of the total classes). It thus requires very few annotated segmentation masks relative to the full dataset. However, the method also results in decent performance with automatic segmentations obtained by a recent method called TokenCut (CVPR '22).

The experiments make sense, particularly the choice of evaluation benchmarks as well as the selection of competing methods. I like that several architectural variants & differently pre-trained instances thereof were considered, as well as repeated runs with different random seeds (see Appendix). I do have some concerns/questions with the experiments however which I list in the next section of the review. I think an opportunity was missed to conduct a more detailed ablation, and I have concerns about the hyperparameter selection procedure for the competing methods. I am surprised that increasing the amount of fine-tuning data (beyond 3*500 images) can harm performance (see Fig. 8).

In terms of clarity, I think the description of the relevance map extraction procedure (mostly relegated to the appendix) could be improved. No need to reproduce the full description from the original paper which describes it, but some clarification is necessary, e.g. I am not sure how this is initialised to an identity map? Are there any relevant knobs and dials to tweak the results?

Overall, I am leaning towards tentative acceptance because I like the method and the direction. The paper makes an important point (made elsewhere too), namely that focusing on a "single measuring stick: [accuracy]" can only get us so far. Instead -- and this is my read -- we have to look into encouraging better model behaviour through other means, e.g. in this case making sure the model attends to relevant parts of the object. Before reaching a final rating, I would appreciate responses to the questions I raise in the rebuttal regarding the experiments.

One unstated assumption that could perhaps be interrogated a bit more is that reliance on background is by default something to be discouraged. While this probably is the case for most classes, there are other classes with more ambiguous appearance where the background is presumably useful. A more fine-grained analysis of the impact on various classes would have been quite useful. This issue is alluded to in section 5 but without any corresponding qualitative or quantitative analysis as far as I can tell.

There is partially such an analysis in the appendix (section H), which compares performance on the 500 training classes. vs the 500 remaining classes. These are interesting results which are briefly discussed in the main paper (section 5). There is an accuracy difference between the two sets, but it is not consistent across models and datasets. Accuracy for the training classes is on average better, but in some cases it is the other way around. Are there any ideas as to why this is the case?

---

> ### Author Response · Authors · 2022-07-28
> **Authors response (part i out of ii)**
>
>
> We thank reviewer Tjbv for the very comprehensive and positive review and for the useful comments and points for discussion. We highly appreciate the attention to detail and the consideration of experiments in the appendices.
>
> Below are our answers to the reviewer’s questions. We would be happy to answer any further questions the reviewer may have.
>
> Please note that all references to the text refer to the revised version of the paper.
>
> __Re. fine-tuning models with the classification loss__
>
> We kindly note that all the models we experiment with have been fine-tuned on ImageNet-1k, which is also the dataset we use for our relevance-based fine-tuning, i.e. the models were fine-tuned with the classification loss (Eq. 10 in the revision, cross-entropy with the ground truth class).
> This is indeed an important clarification, as some of the models were pre-trained on other datasets (e.g. ImageNet-21k), and the phrasing in the main text can be misleading. We thank the reviewer for bringing this to our attention. The revised version has been modified to emphasize this point (L. 127-128).
>
> ---
>
> __Re. hyperparameters selection for baselines__
>
> The difficulty with the hyperparameter search for the baselines stems from the relevance loss term (for the baselines: $\mathcal{L}\_{\text{bg}}$). The goal of the baseline methods is to reduce the relevance values in the background. Therefore it is crucial that we search for hyperparameters that cause a decrease in $\mathcal{L}\_{\text{bg}}$. Thus, for each baseline run, we needed to ensure that the background loss was decreasing (otherwise, the baselines wouldn’t be able to trigger a change in the salient behavior).
> To allow for a fair comparison, we ran a grid search for each model independently to ensure that $\mathcal{L}_{\text{bg}}$ is indeed decreasing through the fine-tuning process.
>
> We kindly refer the reviewer to L. 599-603, where we propose a possible explanation for the instability of this loss in the baseline methods. [Liu et al. Rethinking Attention-Model Explainability through Faithfulness Violation Test. ICML 2022] evaluates different explainability methods for Transformer-based models and has found that vanilla input gradients (as used in the InputxGradient method) violate faithfulness, i.e., do not loyally reflect the salient behavior of the models. For this reason, it is difficult to find hyperparameters to control their values, as they are not necessarily indicative of the network’s relevance and can sometimes even appear to be random.
>
> We note that this behavior is model-dependent. For example, DeiT models were much easier to grid search, and the decrease in $\mathcal{L}\_{\text{bg}}$ was noticeable for various learning rate choices, while ViT AugReg was very difficult to grid search. As can be seen from the TensorBoard training logs for DeiT and ViT AugReg with the same choice of $\lambda_{\text{bg}}$, $\lambda\_{\text{classification}}$: https://imgur.com/a/5X0xuwh, while $\mathcal{L}\_{\text{bg}}$ converges for DeiT with those hyperparameters, this is not the case for ViT AugReg.
>
> ---
>
> __Re. sensitivity tests accuracy (Fig. 8 in Appendix G)__
>
> Our sensitivity tests were conducted with the exact same hyperparameter choice as the main paper, for a fair and unbiased comparison. This, however, may be sub-optimal to some choices of the number of classes and the number of training samples per class. Given more training data, there are more update steps. Therefore, we hypothesize that perhaps a different learning rate scheduler or a slightly lower learning rate would remedy the small drop in accuracy, and possibly even improve the robustness further.
>
> We note that this drop in accuracy is much less significant than the increase in accuracy compared to the baseline method, and that, as the reviewer pointed out, it is evident specifically for ImageNet-A, but less for the other robustness datasets.
>
> Regarding the layers most influenced by the fine-tuning process, we find that in general, the final attention layer is typically the most indicative of the relevance values. This is supported by an ablation done in [Chefer et al. Transformer Interpretability Beyond Attention Visualization. CVPR 2021] which shows that calculating relevance using the last layer is almost equivalent to propagating the relevance throughout the entire network. Accordingly, since our main objective is to fine-tune the relevancy maps, we find that the last attention block is most influenced by the changes.

---

> > ### Author Response · Authors · 2022-07-28
> > **Authors response (part ii out of ii)**
> >
> >
> > __Re. additional ablation tests__
> >
> > Per the reviewer’s request, we have added the proposed ablations to Table 12 in Appendix I. Kindly note that when removing the classification loss, the model is at risk of mode collapse since $\mathcal{L}\_{\text{bg}}$ encourages the relevance on the background to be 0. The mode collapse, in this case, would be to zero out the relevance of the entire image. In an analog manner, $\mathcal{L}\_{\text{fg}}$ encourages a high relevance in the foreground and can cause a mode collapse where all the image receives a relevance of 1. When applied together (the “w/o $\mathcal{L}_{\text{classification}}$ (Eq. 4)” ablation in Table 4), the two losses balance each other. However, when only employing one loss without the other and not adding a regularization term, the fine-tuning would encourage a mode collapse leading to lower accuracy. Due to the current 9-page limit, the results are enclosed in Appendix I and will be moved to the main text for the camera-ready version (which will have an additional page).
> >
> > ---
> >
> > __Re. vanilla ViT vs. ViT AugReg__
> >
> > Loosely rephrased from the AugReg paper- ViT AugReg aims to find the correct balance between the amount of training data, the model size, and “AugReg” (augmentations and regularization) since ViT models rely on AugReg more than CNNs.
> > The authors of explain that this is due to weaker inductive biases for ViTs. By carefully studying those relations, they are able to train models using the public ImageNet-21k that obtain similar performance to similar models that were trained on a much larger dataset (JFT-300M).
> >
> > We thank the reviewer for highlighting this point since we find the experiments on AugReg highly important. These experiments demonstrate that even in the presence of near-perfect augmentations and regularization, our method is still necessary to boost robustness, i.e., augmentations and regularization are not enough to ensure model robustness.
> >
> > ---
> >
> > __Re. background relevance + fine-grained per-class analysis__
> >
> > Regarding the reviewer’s question as to the impact of the background, we concur that the background can be a useful cue, as long as it is not assigned a higher relevance than the foreground. Our goal is not to eliminate the background relevance entirely, but rather to ensure that the relevance on the foreground is higher. For an in-depth, per-class analysis of the impact of our method on each class separately, please see Appendix J, which was attached in the submitted supplementary materials zip, and is now part of the revised pdf file.
> >
> > ---
> >
> > __Re. explainability method description__
> >
> > Following the reviewer’s questions, we have revised Appendix B of the supplementary material to further clarify our use of the GAE method. We would happily add more clarifications upon request.

---

### Meta-Review · Area_Chair_V95X · 2022-08-25

**Recommendation:** Accept
**Confidence:** Certain

**Metareview:**

Initially, this paper received positive reviews. The rebuttal addresses the remaining concerns. All reviewers feel that the contributions of this work are sufficient to merit its acceptance. The area chair agrees with the reviewers and recommends it be acecpted at this conference.

**Award:**

No

---

### Decision · Program_Chairs · 2022-09-14

Accept